# Engrafted parenchymal brain macrophages differ from microglia in transcriptome, chromatin landscape and response to challenge

Anat Shemer[1], Jonathan Grozovski[1], Tuan Leng Tay [2,3,4], Jenhan Tao [5], Alon Volaski[1], Patrick Süß[3], Alberto Ardura-Fabregat[3], Mor Gross-Vered[1], Jung-Seok Kim[1], Eyal David[1], Louise Chappell-Maor[1], Lars Thielecke[6], Christopher K. Glass[5], Kerstin Cornils[7], Marco Prinz[3,8,9] & Steffen Jung [1]

Microglia are yolk sac-derived macrophages residing in the parenchyma of brain and spinal cord, where they interact with neurons and other glial. After different conditioning paradigms and bone marrow (BM) or hematopoietic stem cell (HSC) transplantation, graft-derived cells seed the brain and persistently contribute to the parenchymal brain macrophage compartment. Here we establish that graft-derived macrophages acquire, over time, microglia characteristics, including ramified morphology, longevity, radio-resistance and clonal expansion. However, even after prolonged CNS residence, transcriptomes and chromatin accessibility landscapes of engrafted, BM-derived macrophages remain distinct from yolk sac-derived host microglia. Furthermore, engrafted BM-derived cells display discrete responses to peripheral endotoxin challenge, as compared to host microglia. In human HSC transplant recipients, engrafted cells also remain distinct from host microglia, extending our finding to clinical settings. Collectively, our data emphasize the molecular and functional heterogeneity of parenchymal brain macrophages and highlight potential clinical implications for HSC gene therapies aimed to ameliorate lysosomal storage disorders, microgliopathies or general monogenic immuno-deficiencies.

[1] Department of Immunology, Weizmann Institute of Science, Rehovot 76100, Israel. [2] Cluster of Excellence BrainLinks-BrainTools, University of Freiburg, 79110 Freiburg, Germany. [3] Institute of Neuropathology, Medical Faculty, University of Freiburg, 79106 Freiburg, Germany. [4] Institute of Biology I (Zoology), Faculty of Biology, University of Freiburg, 79104 Freiburg, Germany. [5] Department of Cellular and Molecular Medicine, University of California, San Diego, 9500 Gilman Drive, La Jolla, CA 92093-0651, USA. [6] Institute for Medical Informatics and Biometry, Faculty of Medicine Carl Gustav Carus, Technische Universität Dresden, Dresden, Germany. [7] University Medical Center Hamburg-Eppendorf, Department of Pediatric Hematology and Oncology, Division of Pediatric Stem Cell Transplantation and Immunology and Research Institute, Children's Cancer Center Hamburg, 20246 Hamburg, Germany. [8] BIOSS Centre for Biological Signalling Studies, University of Freiburg, 79104 Freiburg, Germany. [9] CIBSS Centre for Integrative Biological Signalling Studies, University of Freiburg, 79104 Freiburg, Germany. These authors contributed equally: Anat Shemer, Jonathan Grozovski, Tuan Leng Tay. Correspondence and requests for materials should be addressed to S.J. (email: s.jung@weizmann.ac.il)

Macrophages were shown in the mouse to arise from three distinct developmental pathways that differentially contribute to the respective tissue compartments in the embryo and adult. Like other embryonic tissue macrophages, microglia first develop from primitive macrophage progenitors that originate in the mouse around E7.25 in the yolk sac (YS), are thought to be independent of the transcription factor (TF) Myb, and infiltrate the brain without monocytic intermediate[1–3]. YS macrophage-derived microglia persist throughout adulthood. Most other tissue macrophages are however replaced shortly after by fetal monocytes that derive from myb-dependent multipotent erythro-myeloid progenitors (EMP) that also arise in the YS, but are currently thought to be consumed before birth. Starting from E10.5, definitive hematopoiesis commences with the generation of hematopoietic stem cells (HSC) in the aorto–gonado–mesonephros (AGM) region. HSC first locates to the fetal liver but eventually seeds the bone marrow (BM) to maintain adult lymphoid and myeloid hematopoiesis. Most EMP-derived tissue macrophage compartments persevere throughout adulthood without significant input from HSC-derived cells. In barrier tissues, such as the gut and skin, as well as other selected organs, such as the heart, HSC-derived cells can however progressively replace embryonic macrophages involving a blood monocyte intermediate[4].

Differential contributions of the three developmental pathways to specific tissue macrophage compartments seem determined by the availability of limited niches at the time of precursor appearance[5]. In support of this notion, following experimentally induced niche liberation by genetic deficiencies, such as a Csf1r mutation, irradiation, or macrophage ablation, tissue macrophage compartments can be seeded by progenitors other than the original ones[6–9].

Tissue macrophages display distinct transcriptomes and epigenomes[10,11], that are gradually acquired during their development[12,13]. Establishment of molecular macrophage identities depends on the exposure to tissue-specific environmental factors[4,14]. Accordingly, characteristic tissue macrophage signatures, including gene expression and epigenetic marks, are rapidly lost upon ex vivo culture, as best established for microglia[11,15].

Microglia have been recognized as critical players in central nervous system (CNS) development and homeostasis[16]. Specifically, microglia contribute to synaptic remodeling, neurogenesis, and the routine clearance of debris and dead cells[17–21]. Microglia furthermore act as immune sensors and take part in the CNS immune defense[22]. Deficiencies affecting intrinsic microglia fitness can result in neuropsychiatric or neurologic disorders[23]. Therapeutic approaches to these "microgliopathies" could include microglia replacement by wild-type (WT) cells. Moreover, microglia replacement by BM-derived cells has also been proposed as treatment for metabolic disorders, such as adrenoleukodystrophy (ALD) and Hurler syndrome, as well as neuroinflammatory diseases (e.g., amyotrophic lateral sclerosis, Alzheimer's) in order to slow down disease progression or improve clinical symptoms[24]. HSC gene therapy was shown to arrest the neuroinflammatory demyelinating process in a gene therapy approach to treat metachromatic leukodystrophy (MLD) albeit with delay[25]. Of note, replacement of YS-derived microglia by HSC-derived cells is also a by-product of therapeutic stem cell transplantations that are routinely used to treat monogenic immune disorders, such as Wiskott–Aldrich syndrome (WAS) and IL-10 receptor deficiencies. To what extent HSC-derived cells can replace the host microglia (especially after conditioning) and if these restore functions by cross-correction remains unclear. Understanding how engrafted cells perform in the host, in particular following challenge, is therefore of considerable clinical importance, not only in HSC transplantation but also in HSC gene therapy approaches of disorders with a neurological phenotype.

Here we report a comparative analysis of YS-derived microglia and BM graft-derived parenchymal brain macrophages. Using RNAseq and ATACseq of host and engrafted macrophages isolated from mouse BM chimeras, we show that these cells acquire microglia characteristics such as longevity, morphology, and gene expression features, but still remain significantly distinct with respect to transcriptomes and chromatin accessibility landscapes. Furthermore, host and graft cells display discrete responses to challenge by peripheral endotoxin exposure. Finally, by extending our finding to clinical settings, we confirm that in human HSC transplant patients, grafted cells also remain distinct from host microglia. Collectively, these data establish that engrafted macrophages differ from host microglia even after prolonged residence in the brain parenchyma and could have considerable clinical implications for patients treated by HSC gene therapy.

## Results

**Engrafted brain macrophages acquire longevity and radioresistance.** Following total body irradiation (TBI), myeloid precursors enter the brain and contribute to the parenchymal macrophage compartment[26–28]. Host microglia are relatively radio-resistant and unless combined with conditioning, engraftment of the brain macrophage pool was therefore reported to be limited[27,29–31]. It furthermore remained unclear to what extent graft-derived cells acquire over time microglia characteristics, such as longevity and radioresistance. To address this issue, we generated BM chimeras by lethally irradiating WT mice (CD45.2) (950 cGy) and transplanting them with $CX_3CR1^{GFP}$ BM (CD45.1)[32] (Supplementary Fig. 1a). Four months after irradiation and BM transfer (BMT), monocyte precursors in the BM and circulating blood monocytes of the chimeras were all CD45.1$^+$ GFP$^+$ and hence exclusively derived from the BM graft (Supplementary Fig. 1b, c). Analysis of the CD45$^{int}$ CD11b$^+$ Ly6C$^-$ Ly6G$^-$microglia compartment of the chimeras revealed the presence of GFP$^+$ graft- and GFP$^-$ host-derived cells (Supplementary Fig. 1d). In line with earlier reports, grafted cells initially constituted only a small fraction of the parenchymal brain macrophage population. However, the cells progressively replaced the host microglia (Supplementary Fig. 1d). The expansion of the GFP$^+$ infiltrate could indicate ongoing peripheral input. Alternatively, the undamaged nonirradiated CNS-resident graft-derived cells might have an advantage over the irradiated host microglia, and gradually outcompete the latter during the reported infrequent microglial local proliferation[33,34]. To distinguish between these options, we performed a tandem engraftment. Recipient mice (CD45.1/2) were irradiated twice, 15 weeks apart, followed by engraftment with $CX_3CR1^{GFP}$ BM (CD45.1) and $CX_3CR1^{Cre}$:R26-RFP$^{fl/fl}$ BM (CD45.2), respectively (Fig. 1a). Blood analysis of the chimeras 7 weeks after the second BMT, showed that monocytes were exclusive derivatives of the second graft (Fig. 1b), as were myeloid BM precursors (Supplementary Fig. 1e). In contrast, analysis of the CNS compartment of the chimeras revealed the presence of three distinct macrophage populations: host microglia (CD45.1/2$^+$), cells derived from the first graft (CD45.1$^+$ GFP$^+$), and cells derived from the second BM graft (CD45.2$^+$ RFP$^+$) (Fig. 1c, d, Supplementary Fig. 1f). The presence of cells derived from the first graft establishes that these (1) acquired radioresistance and (2) persisted in the chimeras for more than 2 months without contribution from the periphery. Corroborating the above results, cells derived from both grafts expanded on the expense of the host microglia (Fig. 1d). $CX_3CR1^{GFP}$ (CD45.1) and $CX_3CR1^{Cre}$:R26-RFP$^{fl/fl}$

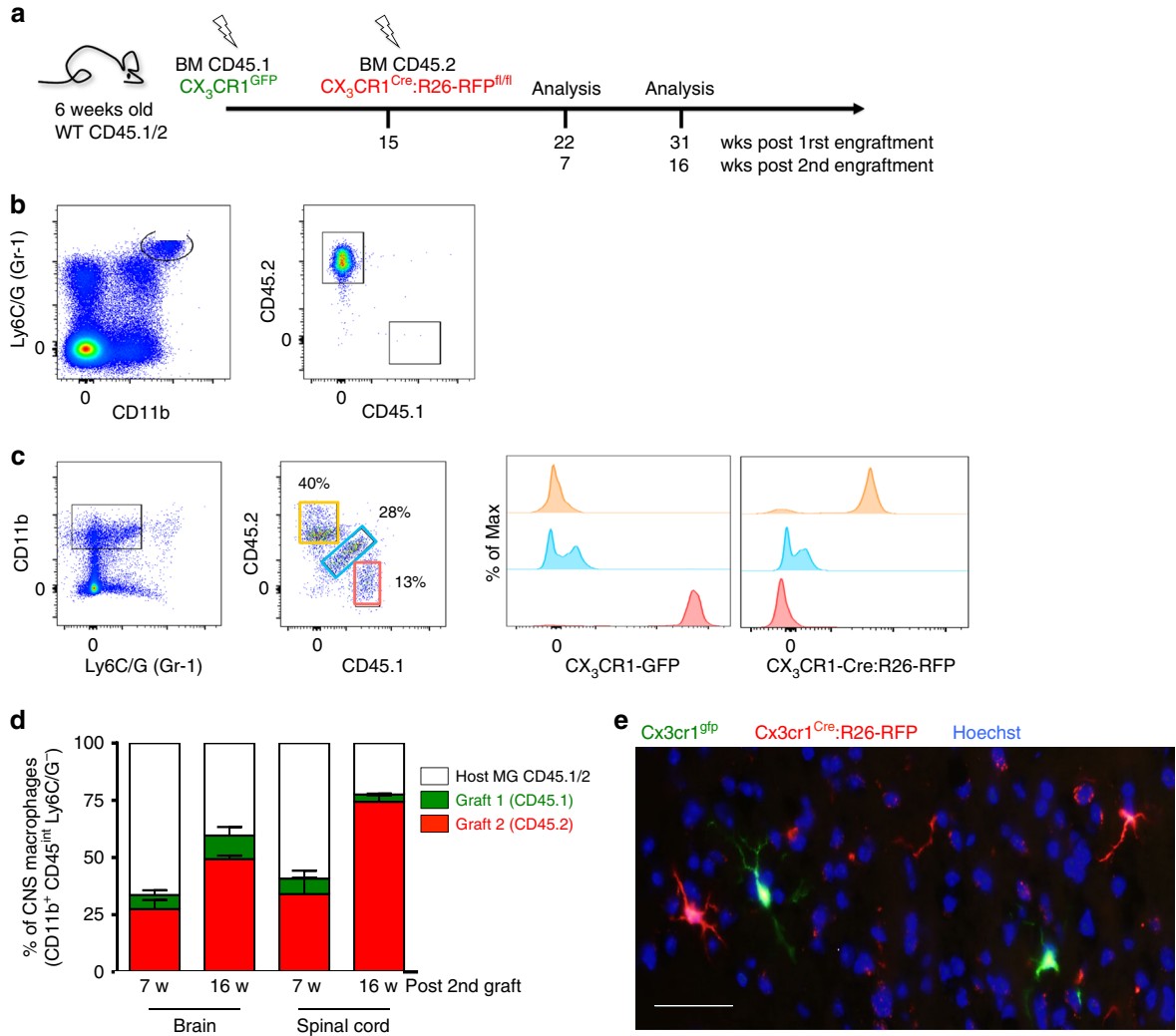

**Fig. 1** Engrafted brain macrophages accumulate over time and self-maintain. **a** Schematic of tandem BM transfer protocol. **b** Flow cytometric blood monocyte analysis of chimera 16 weeks post second transplantation. **c** Flow cytometric analysis of myeloid brain cells 16 weeks post second BMT revealing host microglia (CD45.1/2+, blue), cells derived from the first graft (CD45.1+ GFP+, red) and cells derived from the second graft (CD45.2+ RFP+, orange). **d** Distribution of host and graft-derived cells out of the total Ly6C/G (Gr1)- CD45lo CD11b+ cells in brain and spinal cord at two time points. Data are a summary of six mice. **e** Histological analysis 7 weeks post second BMT revealing ramified GFP+ and RFP+ cells with microglia morphology (scale bar 20 μm). Representative picture

(CD45.2) cells with ramified microglia morphology were detected in the brain parenchyma (Fig. 1e). Collectively, these data establish that engrafted cells adopt microglia characteristics, such as relative radioresistance, longevity, and morphology.

**Macrophages engrafting conditioned brains display expansion.** To further characterize the engraftment process, including clonal dynamics of the BM-derived cells, we used two complementary approaches, comprising (1) transplantation of BM isolated from *Cx3cr1CreER:R26Confetti* ("Microfetti") mice[34] (Fig. 2a) and (2) transplantation of lineage-negative BM cells marked by a genetic barcode prior to transplantation[35] (Supplementary Fig. 2a).

*Cx3cr1CreER:R26Confetti* BM recipients were treated at 2 or 10 weeks post engraftment with tamoxifen (TAM) to induce stochastic expression of one of the four fluorescent reporter proteins encoded by the Confetti construct[34] (Fig. 2a). Thirty weeks later, distinct brain regions of the chimeras, including the olfactory bulb, cortex, hippocampus, and cerebellum, were analyzed for the presence of graft-derived labeled cells and clonal

clusters (Fig. 2b–f). Engraftment was seen in all brain regions analyzed (Fig. 2g), although the cerebellum displayed higher frequencies of BM-derived cells, as reported earlier[29]. Integration of engrafted cells into the endogenous microglial network was reflected by their Iba-1 expression, similar morphology, and intercellular distances as compared to host microglia (Fig. 2c–f, Supplementary Fig. 3). Absence of Confetti+ monocytes over 10 weeks after BM transplantation excluded sustained labeling of peripheral monocytes, indicating that the Confetti-labeled macrophages observed in the 10-week TAM treatment group were derived from the initial engraftment. Single cells or clones (defined as ≥ two same-color cells with 100-μm nearest-neighbor proximity) of Confetti-labeled cells were observed at similar frequency, regardless of the timing of TAM treatment in the olfactory bulb, cortex, and hippocampus (Fig. 2g–i), suggesting comparable proliferation capacities after engraftment. The higher number of cerebellar engrafted macrophages and clones in the 10-week TAM treatment group (Fig. 2j) might be attributable to niche-dependent differences in kinetics of clonal expansion[34]. Clusters of same-color Confetti-labeled grafts in close proximity

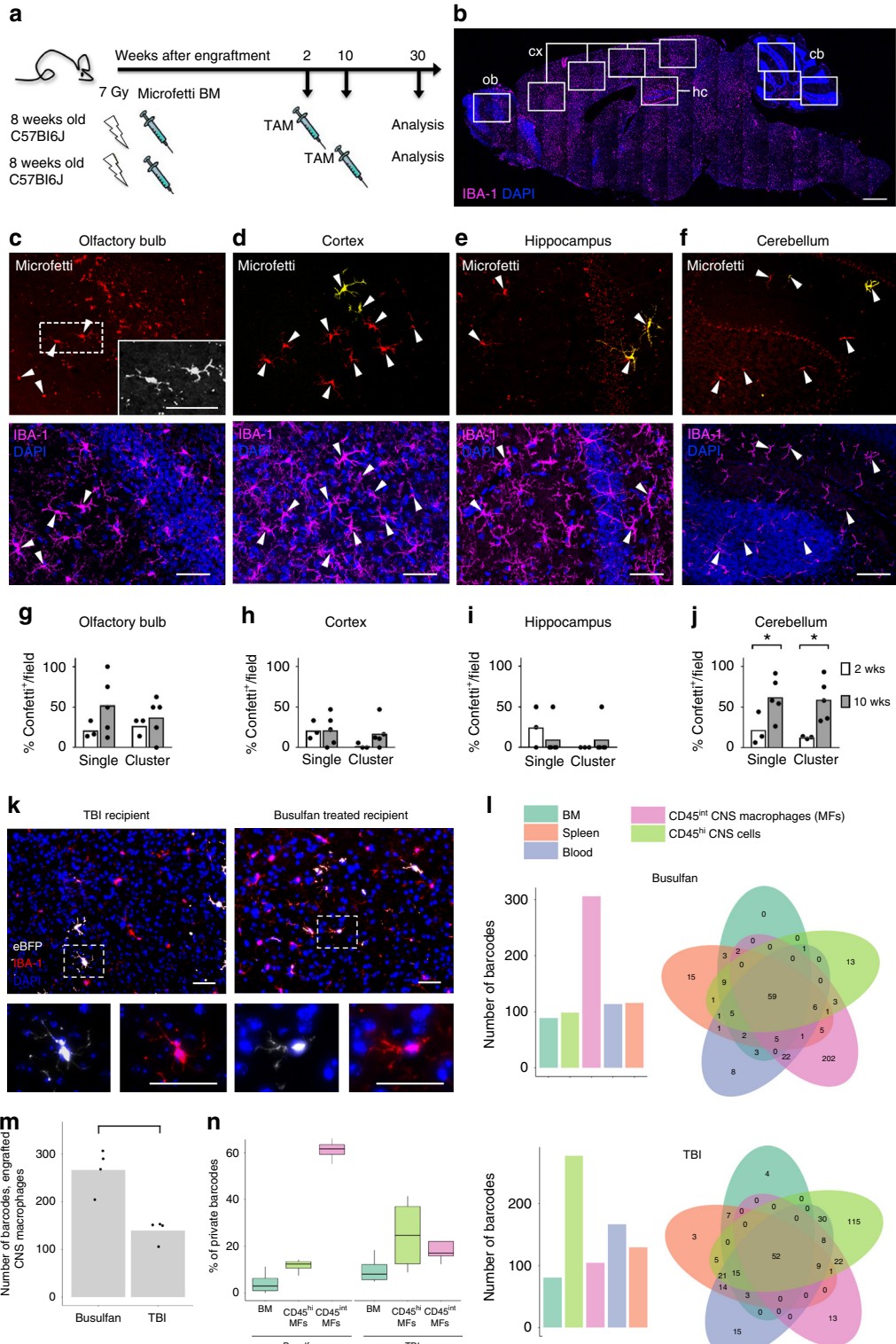

provide evidence of local proliferation, since slow-renewing cortical microglia in 'Microfetti" brains do not form clones over 36 weeks[34].

For the genetic barcoding procedure, lineage-negative BM cells isolated from male donors (CD45.1) were transduced with lentiviral vectors harboring advanced genetic barcodes (BC32) and a BFP reporter gene[35,36]. Transduced HSC/HPCs were transferred into 8-week-old female recipient mice (CD45.2) that were irradiated (950 cGy) or conditioned with the alkylsulfonate

busulfan (BU) (125 mg) used for myeloablation in pediatric and adult patients[37], as well as preclinical mouse models[38,39]. Six months after transplantation, the right hemisphere of the mice was used for immunohistochemistry (eBFP, Iba-1; Fig. 2k). From the left hemisphere, macrophages were sorted and analyzed for presence and complexity of barcodes using bioinformatics[36]. Both conditioning protocols resulted in efficient engraftment of the periphery and the microglia compartment (Supplementary Fig 2b, c), as also reflected in the numbers of clones displaying unique

**Fig. 2** Engrafted brain macrophages display clonal expansion. **a** Fate-mapping scheme for donor "Microfetti" BM cells in lethally irradiated WT hosts. A single dose of TAM was applied at 2 or 10 weeks after BM reconstitution. BM chimeras were analyzed at 30 weeks after BM transplantation. **b** Representative sagittal brain section indicating the fields of view (rectangles) analyzed for IBA-1 (magenta) expressing graft microglia in the olfactory bulb (ob), cortex (cx), hippocampus (hc), and cerebellum (cb). DAPI (blue). Scale bar, 1 mm. **c–f** Representative images of Confetti$^+$ (yellow/ red) Iba-1$^+$ (magenta) donor cells (arrowheads) in the (**c**) olfactory bulb (higher magnification in grayscale, inset), (**d**) cortex, (**e**) hippocampus, and (**f**) cerebellum. DAPI (blue). Scale bars, 50 μm (**c–e**) and 100 μm (**f**). **g–j** Frequency of Confetti$^+$ IBA-1$^+$ graft as single cells or clones in analyzed fields from TAM treatment groups of 2 (white) and 10 weeks (gray) after BM transplantation. Each dot represents the mean quantification of one animal. At least four sections per animal were analyzed. Mann–Whitney test, *$P = 0.0357$, 0.0179, respectively in (**j**). **k** Representative images of cortical eBFP$^+$ (white) IBA-1$^+$ (red) graft-derived cells in brains of TBI and busulfan-conditioned mice. DAPI (blue). Higher magnification in insets. Scale bars, 50 μm. **l** Representative barcode analysis on DNA extracted from peripheral blood, bone marrow, spleen, and sorted CD45$^{int}$ brain macrophages and CD45$^{high}$ cells of a busulfan- and a TBI-conditioned mouse. Barcode numbers for each sample are shown in the bar plot and the amount of shared barcodes per sample are displayed in the Venn diagrams. **m** Number of barcodes detected in engrafted macrophage samples of TBI- and busulfan- conditioned animals. Each dot represents one animal ($n = 4$ per group). **n** Bar graph showing percentages of barcodes private to grafted cells among CD45$^{int}$ and CD45$^{hi}$ CNS cells (i.e., parenchymal and non-parenchymal macrophages, respectively), as well as BM cells of TBI- and busulfan-conditioned animals ($n = 4$ per group)

barcodes (Fig. 2l, m, Supplementary Fig. 2d). Interestingly, while engrafted cells isolated from BM, blood, and spleen, and also CD45$^{hi}$ CNS cells representing hematopoietic non-microglial cell, displayed mainly shared clones, the majority of clones identified among brain macrophages of busulfan-conditioned animals did not have counterparts in other tissues and were private (Fig. 2n). Corroborating our above results and that of others[31,40], this suggests that a major fraction of the grafted cells originates from transduced precursors that seed the host CNS early after engraftment and is maintained by local proliferation independent from ongoing hematopoiesis. Taken together, these results establish that efficiently engrafted donor cells adopt the phenotype and distribution of resident microglia within this cellular network and expand locally with kinetics specific to brain regions and conditioning paradigms.

**Engrafted cells have transcriptomes distinct from microglia.** BM graft-derived parenchymal brain macrophages acquire characteristics such as ramified morphology, longevity, and radioresistance and can hence be considered engrafted microglia-like cells that could potentially be employed for therapeutic purposes. Recent studies have highlighted the impact of the tissue environment on macrophage identities, including epigenomes and expression profiles[10,11]. To test whether graft-derived microglia acquire in the CNS such a global molecular imprint, we transplanted lethally irradiated 6-week-old WT mice with congenic WT BM harboring CD45.1 alleles. Nine months post-transplantation, chimeras were sacrificed and brain macrophages were isolated for transcriptome and epigenome analysis by RNAseq[41] and ATACseq[42], respectively. At this time point, half of the CNS macrophages of the chimeras were of graft origin (Fig. 3a, Supplementary Fig. 4a).

Global RNAseq analysis of parenchymal host and graft brain macrophages isolated from individual BM chimeras revealed that engrafted cells and host microglia showed significant transcriptome overlap, clearly distinguishing them from monocytes that served as reference for a HSC-derived cell population[43] (Fig. 3b, c). Of the total 11,614 detected transcripts, 10,635 (91%) displayed a lower than twofold difference between the engrafted cells and host microglia. On the other hand, 979 transcripts were differentially expressed between these two populations (absolute value of log2-fold change >1, p-value < 0.05) (Supplementary Fig. 5a). Expression of 469 genes was restricted to host microglia, while 510 genes were uniquely expressed by the engrafted cells (Supplementary Fig. 5a, b). Engrafted macrophages, but not host microglia displayed, for instance, mRNA encoding CCR2, lysozyme, CD38, CD74, Mrc1, ApoE, and Ms4a7 (Fig. 3e, Supplementary Fig. 5b). Differentially expressed genes included transcription factors (TFs) such as the basic helix–loop–helix TF

Hes1, the Krueppel-like zinc finger TF Klf12, the retinoic acid receptor RxRg, and the TGFβ-associated signal transducer Smad3, were preferentially transcribed in host cells. Conversely, engrafted macrophages displayed increased expression of the estrogen receptor Esr1, the runt-domain TF Runx3, and the macrophage survival factor Nr4a1, as compared to host cells (Fig. 3d). Of note, the host microglia in this case were irradiated and differences observed between the two populations could hence arise from radiation damage; transcriptomes of engrafted cells however also differed significantly from age-matched non-irradiated microglia (Supplementary Fig. 5c).

Supporting the notion of their microglia-like identity, engrafted cells expressed similar levels of the DNA–RNA binding protein TDP-43 encoded by the *Tardbp* gene recently implied as a regulator of microglial phagocytosis[44], the two-pore domain potassium channel THIK-1, encoded by the *Kcnk13* gene, and shown to be critical for microglial ramification, surveillance, and IL-1b release[45] and the TF Mef2c, reported to restrain microglia responses[46] (Fig. 3f) Likewise, the graft also displayed expression of "microglia signature" genes that have been proposed to distinguish these cells from other tissue macrophages and acute monocyte infiltrates associated with inflammation[47,48], including Fc receptor-like molecule (*Fcrls*) and TGFβ receptor (*Tgfbr*), which is critical to establish microglia identity[47] (Fig. 3f). Other proposed "microglia signature genes", such as *P2ry12, Tmem119, SiglecH*, and *HexB* displayed either significantly reduced expression in the grafted cells or were exclusively expressed by host microglia, like the ones encoding the sodium/glucose cotransporter 1 (*Slc2a5*), the phosphoglycoprotein protein CD34 (*Cd34*), and the transcriptional repressors Sall1 (*Sall1*) and Sall3 (*Sall3*) (Fig. 3g, Supplementary Fig. 5d). Of note, lack of some microglia markers had been reported before for cells retrieved from acutely engrafted brains[8,47,49,50].

The expression signature of the engrafted macrophages showed a considerable overlap with the transcriptome of perivascular macrophages[51–53], including present and absent transcripts, such as *ApoE, Msn4a7, Slc2a5*, and *Sall1*, respectively. Gene set enrichment analysis (GSEA)[52] revealed that engrafted macrophages displayed an activation signature as compared to host microglia (Supplementary Fig. 6a). Finally and corroborating our data, the list of genes we report as differentially expressed by engrafted and host cells also displayed a considerable overlap with results recently reported by two other groups[54,55] (Supplementary Fig. 6b).

Sall1, a member of the *Spalt* ("Spalt-like" (Sall)) family of evolutionarily conserved transcriptional regulators critical for organogenesis, acts as a repressor by recruitment of the nucleosome remodeling and deacetylase corepressor complex (NurD). Binding motifs of Sall1 and hence its direct genomic

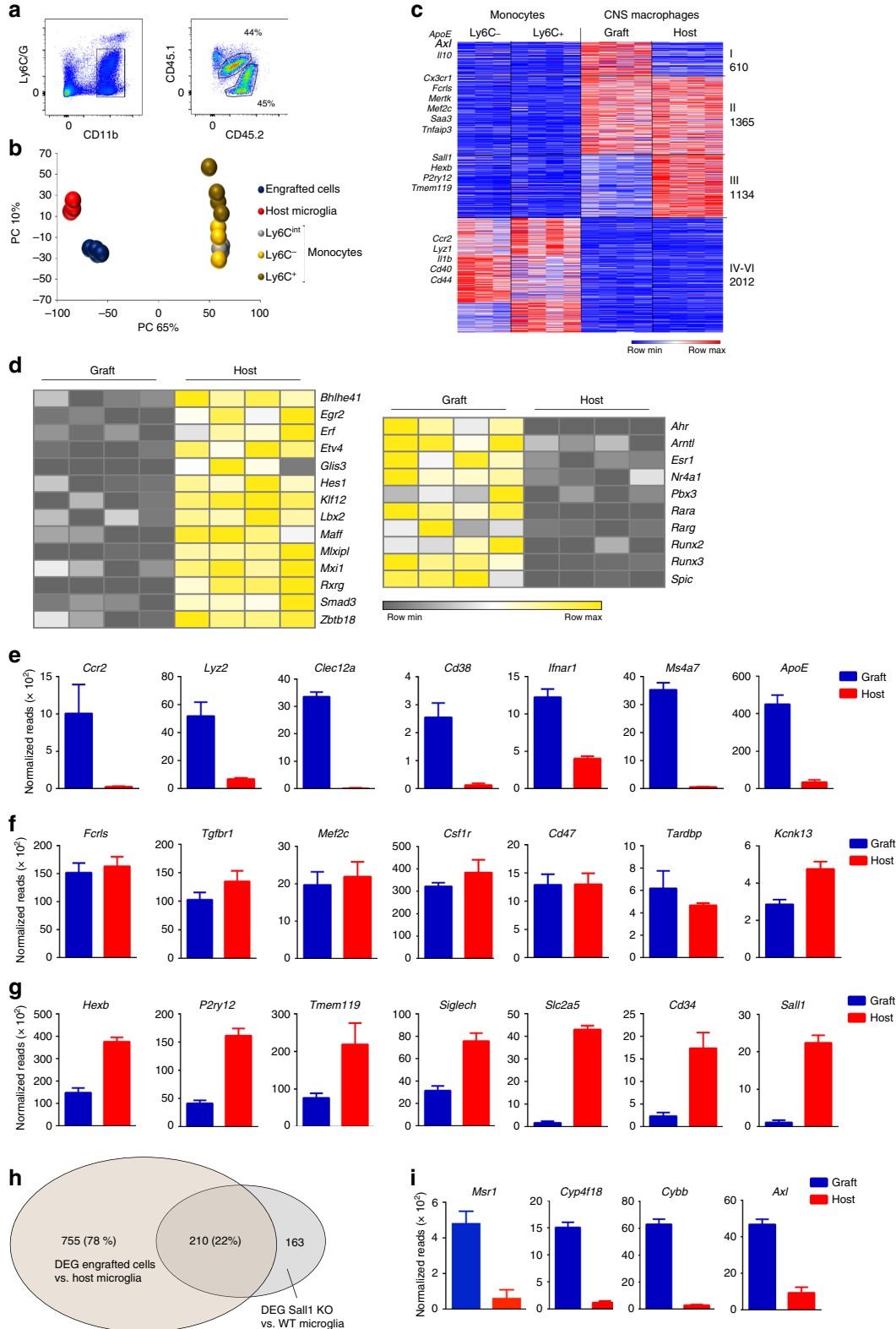

targets remain undefined precluding a direct assessment of the impact of the lack of the repressor on the expression signature of the grafted macrophages. Interestingly though, comparison of the recently reported list of genes differentially expressed by WT and Sall1-deficient microglia[50] and that of host and graft brain macrophages revealed in this study, showed significant overlap (Fig. 3h). This included expression of genes otherwise restricted

to macrophages residing in non-CNS tissues, such as *Msr1*, encoding a scavenger receptor, and *Cyp4f18* encoding cytochrome P450 (Fig. 3i). Furthermore, like Sall1-deficient microglia[50], grafted CNS macrophages displayed an activation signature, as reflected by expression of *Cybb* encoding the Cytochrome b-245 heavy chain and *Axl* encoding a member of a tyrosine kinase receptor family critical for debris clearance[20] (Fig. 3i,

**Fig. 3** Comparative transcriptome analysis of grafted cells and host microglia. **a** Gating strategy for isolation of CNS macrophages. Host microglia were defined as Ly6C/G⁻CD11b⁺CD45.2^lo cells; graft-derived cells were defined as Ly6C/G⁻CD11b⁺CD45.1^lo cells. **b** Principal component analysis of transcriptomes of engrafted cells and host microglia and transcriptomes of monocytes subsets[46]. Source data are provided as a Source Data file. **c** Heatmap of RNA seq data of engrafted cells and host microglia compared to transcriptomes of Ly6C⁻ and Ly6C⁺ monocyte subsets[46]. Analysis was restricted to genes, which showed a twofold difference and yielded p-value < 0.05 between at least two populations. Source data are provided as a Source Data file. **d** Heatmap showing differential TF expression profiles of engrafted cells and host microglia. Source data are provided as a Source Data file. **e** Examples of significantly differential (log2FC>1 and p-value < 0.05) gene expression enriched in engrafted cells. Graphs show normalized reads from RNA seq data of samples acquired in (**a**), n = 4. **f** Examples of genes expressed in similar levels in host and engrafted cells. Graphs show normalized reads from RNA seq data of samples acquired in (**a**), n = 4. None of the genes were differentially expressed (log2FC < 1). *Tgfbr1* and *Kcnk13* were of low significance (p-value < 0.05) but did not meet our FC threshold. **g** Examples of significantly differential (log2FC > −1, p-value < 0.05) gene expression enriched in host microglia. Graphs show normalized reads from RNA seq data of samples acquired in (**a**), n = 4. **h** Venn diagram showing overlap of genes differentially expressed by WT and Sall1-deficient microglia[53], and genes differentially expressed by host and engrafted brain macrophages. **i** Examples of gene expression of engrafted and host cells of genes expressed in non-CNS macrophages or Sall1-deficient microglia. Source data are provided as a Source Data file

Supplementary Fig. 6b). This suggests that a major fraction of the differential expression of host microglia and engrafted cells could be explained by the specific absence of the transcriptional repressor Sall1 from the former cells. Overall, these findings establish that engrafted macrophages that persist in the brain and acquire microglia characteristics such as morphology and radio-resistance, also show significant transcriptome overlap with host microglia, but remain a molecularly distinct population.

**Distinct chromatin accessibility landscapes in grafted and host cells**. To further define engrafted cells and host microglia, we performed an epigenome analysis using ATACseq that identifies open chromatin regions by virtue of their accessibility for "tagmentation" by transposases[42]. Correlated ATACseq replicates (Supplementary Fig 7a) performed on graft and host microglia isolated from BM chimeras detected 58,947 total accessible regions (corresponding to 16,156 genes). Corroborating the observed differential gene expression (Fig. 3), host microglia but not engrafted cells, displayed ATAC signals in the *Sall1* and *Klf2* loci, as indicated in Integrative Genomics Viewer (IGV) tracks (Fig. 4a). ATAC peaks in other genomic locations, such as *ApoE* and *Ms4a7*, were restricted to genomes of engrafted cells, in line with mRNA detection in these cells, but not host microglia (Fig. 4b). As ATACseq does not discriminate between bound transcriptional activators and repressors, some differentially expressed loci did not show epigenetic differences. This included for instance the MHC II locus (*H2-ab1*), which displayed similar ATACseq peaks in host microglia and the graft that lack and display *H2-ab1* transcripts, respectively (Fig. 4c). These loci might be transcriptionally silenced, but activated upon cell stimulation. Similar "poised" states, that might be revealed following challenge, can be assumed for gene loci, that displayed differential ATAC peaks, but were transcriptionally active in neither the host nor the engrafted cells. Finally, rare genes, such as the *Dbi* locus showed equal expression, but differential ATAC profiles suggest that their transcription is driven by distinct TFs (Fig. 4c). Global quantification of differential ATAC peaks between the two brain macrophage populations revealed that 6% of the total accessible regions (or 8.7% of the associated genes) were distinct. Specifically, 1506 peaks (corresponding to 941 genes) displayed a >4-fold significant (p-value < 0.01) enrichment in host microglia and 2176 peaks (corresponding to 1465 genes) were increased in BM graft-derived cells (Fig. 4d).

To identify potential TFs responsible for the differential transcriptomes and chromatin accessibility landscapes of engrafted and host microglia, we applied a TF Binding Analysis (TBA) machine learning model[56]. TBA learns to jointly weigh 100s of motifs drawn from the JASPAR and CISBP databases to distinguish open chromatin sites from GC-matched genomic background. By examining the contribution of each motif to the

model's performance, TBA assigns a significance level to each motif. This analysis revealed a class of highly significant motifs correlated with open chromatin that were common to both host and graft cells and a variety of other myeloid cells (p < 10e−20) such as CTCF, SPI1, and Runx binding motifs (Supplementary Fig. 7b). Of the more intermediately ranked motifs, 41 motifs were more important in either host or graft cells as measured by the log-likelihood ratio (LL) when comparing graft and host microglia (LL >= 10e−2, blue points Fig. 4e). Ten motifs were preferentially detected in host microglia (Fig. 4e), including motifs for TFs such as IRF8, which is a critical regulator of microglia identity[57], as well less characterized motifs such as the C2H2 zinc finger motif. The latter motif could potentially be recognized by Sall1, which contains tandem C2H2 zinc fingers. Motifs preferentially detected in grafted cells included CEBPa, RFX, and Jun motifs (Fig. 4e). Motifs that were preferentially detected in either graft or host cells exhibited even greater differences in comparison to other myeloid cell populations, consistent with environmental cues directing chromatin remodeling of grafted cells (Fig. 4f). Collectively, these observations substantiate the conclusion that while grafted cells adopt epigenetic characteristics similar to microglia, they remain distinct from host cells even after prolonged CNS residence.

**Engrafted macrophages respond unlike microglia to challenge**. Given the significant transcriptome and chromatin landscapes differences between the host and BM graft-derived macrophages that persists for extended periods of time post transplantation, we next examined whether the two populations are functionally distinct. To that end, chimeras were challenged 9 months post-transplantation by a peripheral injection of the bacterial endotoxin lipopolysaccharide (LPS), an established paradigm for inflammation associated with robust microglia responses to systemic cytokine secretion. Host and engrafted cells were isolated from the brains of the chimeras 12 h post LPS challenge, global RNA and ATAC sequencing were performed, and the results were compared to the samples obtained from non-challenged mice presented earlier (Figs. 3, 4). Principal component analysis (PCA) revealed a high degree of similarity within each group, but segregation of the host and graft samples (Fig. 5a). Host and grafted cells responded with altered expression of 745 shared genes. In total, 940 genes were changed in grafted cells only, and 602 genes were changed in host microglia upon LPS challenge (Fig. 5b). Examples for these three categories are shown in Fig. 5c, d and Supplementary Fig. 8a. Genes commonly induced by engrafted cells and host microglia in response to the LPS challenge comprised *Tnf*, *Ccl5,* and *Tnfaip3*, encoding the A20 deubiquitinase that negatively regulates NF-κB-dependent gene expression. Commonly downregulated genes comprised *Trem2*, *Cx3cr1*, and *Aif1*. The graft-specific response included

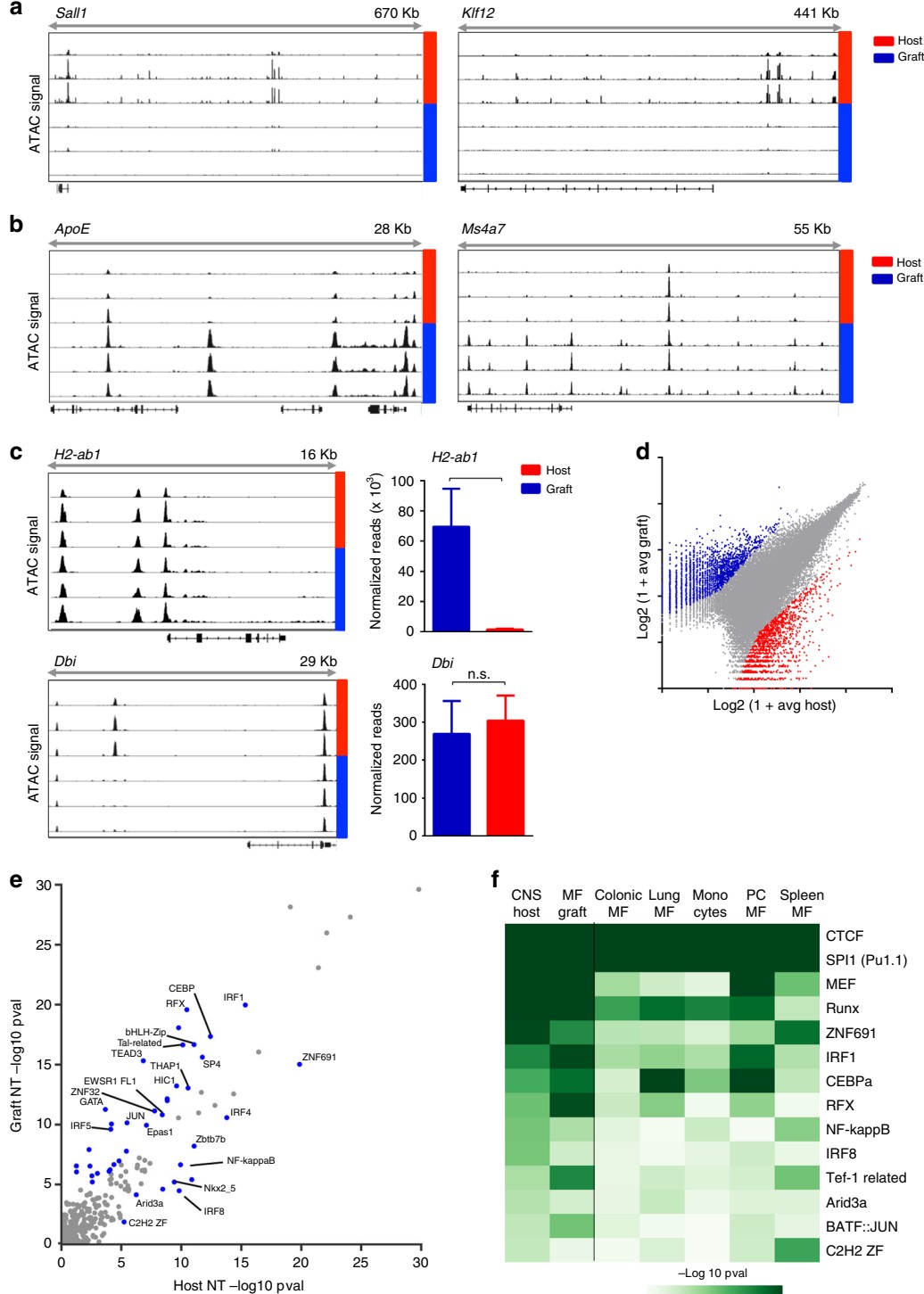

**Fig. 4** Comparative epigenome analysis of grafted cells and host microglia. **a** IGV tracks of *Sall1* and *Klf2* loci showing ATAC signals in host (red) but not engrafted (blue) cells. $N = 3$. **b** IGV tracks of *ApoE* and *Ms4a7* loci showing ATAC signals in engrafted cells (blue) but not host (red) microglia. $N = 3$. **c** ATAC-seq IGV tracks (left, $n = 3$) and normalized RNA seq reads (right, $n = 4$) of *H2-ab1* and *Dbi* in host microglia (red) and engrafted cells (blue). +—$p$-value < $10^{-5}$, n.s.—$p$-value > 0.05. **d** Analysis of all 58,947 detected ATAC peaks, from which 1506 peaks and 2,176 peaks displayed >4-fold significant ($p$-value < 0.01) enrichment in host microglia and engrafted cells, respectively. Source data are provided as a Source Data file. **e** Comparison of motif significance in resting host and grafted cells. *P*-values were calculated using TBA models trained on intergenic peaks from host and grafted cells. Significant motifs that show a large difference ($p < 10e{-}5$, log-likelihood ratio > = 2) are indicated in blue points. Source data are provided as a Source Data file. **f** Heatmap of the significance of motifs in various myeloid cell types and host cells. The intensity of the color indicates greater significance (−log10 *p*-value) of each motif. Source data are provided as a Source Data file

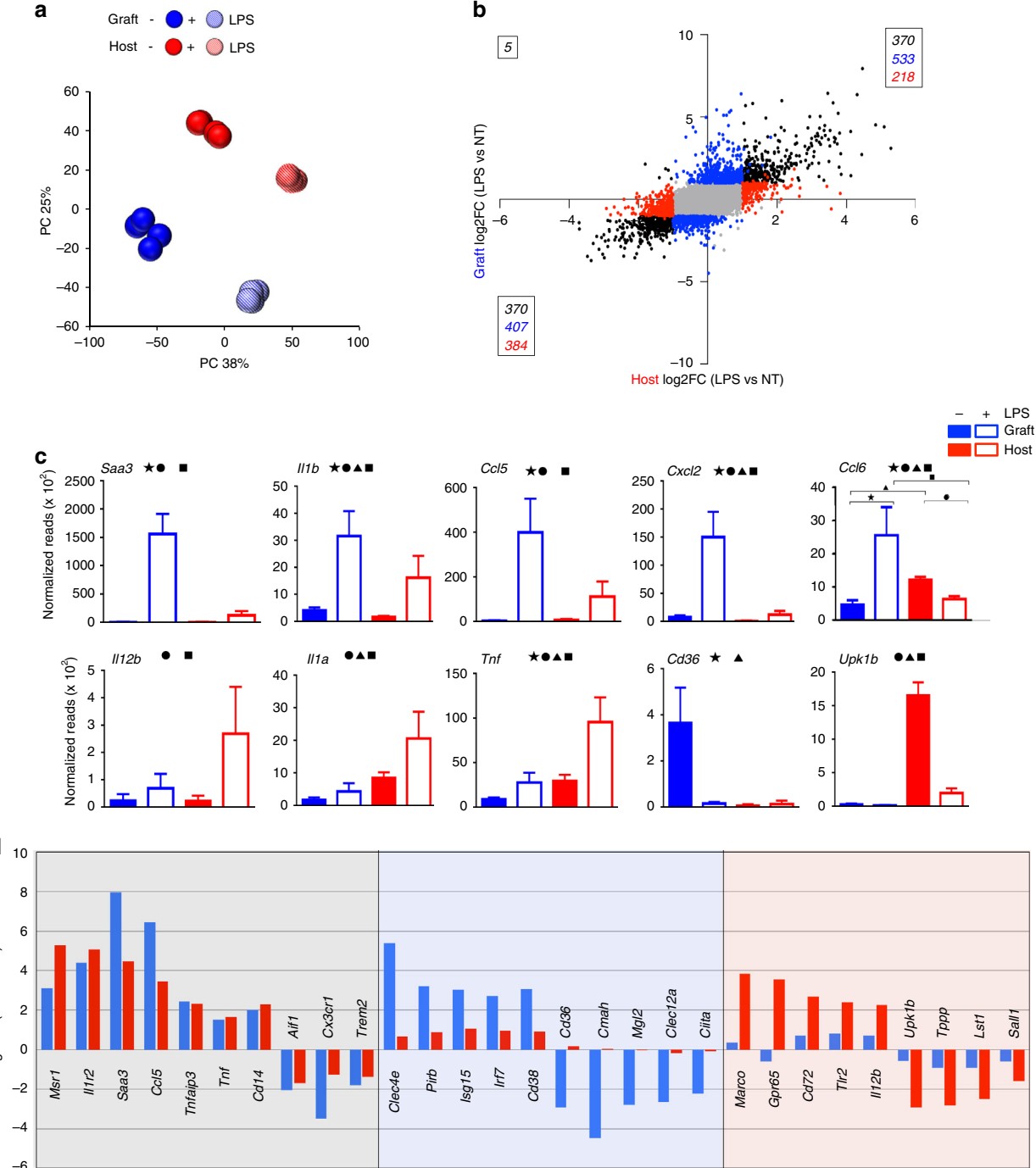

**Fig. 5** Distinct LPS responses of engrafted macrophages and host microglia. **a** PC analysis of RNA seq data of graft and host microglia in steady state and 12 h post LPS. **b** Expression analysis of grafted cells and host microglia in steady state and 12 h post LPS by RNA seq. **c** Examples of genes expression of graft and host microglia in steady state and 12 h post LPS. Significance is indicated by the symbols (*p*-value < 0.05), or lack thereof (*p*-value>0.05). See Ccl6 plot (top right) for symbol to condition conversion. **d** Fold expression change of selected genes significantly different (absolute value of log2FC > 1, *p*-value < 0.05) following challenge in grafted cells (middle) or host microglia (right), as well as genes displaying comparable up- and downregulation in both engrafted cells and host microglia (left). Source data are provided as a Source Data file

upregulation of *Clec4e*, *Pirb*, *Isg15*, and *Irf7* and downmodulation of *Mgl2*, *Cmah*, and *CD36*. Genes specifically induced in host microglia encoded the scavenger receptor Marco, Gpr65, Tlr2, and Il12b. Moreover, host microglia silenced the expression of *Sall1* and *Upk1b* (Fig. 5d). Ingenuity analysis of transcriptomes of engrafted and host brain macrophages isolated from chimeras with and without peripheral LPS challenge revealed potential distinct upstream regulators acting on these populations, as well

as a differential representation of activated functional pathways (Supplementary Table 1). Engrafted macrophages displayed for instance activation of pathways controlled by IL1b and Ifng and suppressed by IL-10.

ATACseq analysis revealed differential epigenome alterations between engrafted and host cells in response to the LPS challenge. Specifically, correlated ATACseq replicates (Supplementary Fig 9a) performed on engrafted and host cells isolated from BM

chimeras after LPS challenge detected 46,485 total accessible regions (corresponding to 15,390 genes). Global quantification of differential ATAC peaks between the two brain macrophage populations revealed a total of 552 peaks (corresponding to 391 genes) that displayed a >4-fold significant ($p$-value < 0.01) enrichment in host microglia and 841 peaks (corresponding to 618 genes) that were increased to the same extent in BM graft-derived cells (Fig. 6a).

Overall, differences between host and graft cells were less pronounced after LPS challenge (Figs. 4d, 6a). Analysis of motifs with TBA models trained on intergenic peaks present after LPS challenge identified a core group of highly significant motifs common to graft and host cells in both naive and LPS conditions that may be important for maintaining microglia identify ($p$ < 10e −20, Supplementary Fig. 9b). Under LPS conditions, we observed that AP-1 family motifs (Jun-related, Fos-related) and NF-kappaB (Rel, Nfkb1) motifs are more significant, which is consistent with the role AP-1 and NF-kappaB factors play in mediating the macrophage inflammatory response (Fig. 6b). Interestingly, host and graft cells preferred different variants of the NF-kappaB and AP-1 motifs (motif logos, Fig. 6b). The NF-kappaB motif most highly enriched in host cells was more GC rich than that in graft cells, whereas the AP-1 motif most enriched in graft cells corresponded to an N(1) spaced motif (TGANTCA) in contrast to the N(2) motif (TGANNTCA) that was most highly enriched in host cells. Of the 27 motifs that were differentially detected in challenged host and graft cells, several motifs were also differentially detected in untreated cells, including motifs for IRF8, zinc finger factors, NK-related factors, and Tal-related factors (Figs. 4f, 6b).

Appearance of ATACseq peaks was correlated with differential gene expression between the two macrophage populations (Fig. 6c). Transcripts encoding the scavenger receptor Marco were absent from host microglia, but specifically induced in these cells but not engrafted cells following the LPS challenge (Fig. 6c). Likewise, the *Marco* locus (92kb) in microglia of unchallenged animals displayed five ATACseq peaks that were all restricted to the host cells (I–V; Fig. 6c). LPS challenge resulted in loss of one peak (IV) and the induction of three additional peaks (VI–VIII), again restricted to host microglia (Fig. 6c). An induced peak located 53,411 bp downstream of the TSS displayed a host-specific Nfkb1 motif, whereas a second peak located 19,210 bp upstream of the Marco TSS displayed a host-specific AP-1 (Fos-related) motif (Fig. 6c). Engrafted macrophages on the other hand, displayed as compared to host microglia prominent induction of secreted phosphoprotein 1 (SPP1)/osteopontin (Fig. 6d), a factor that got recent attention as part of a microglia signature that could be associated with certain CNS pathologies[58]. In accordance with the expression results, *Spp1* loci (74 kb) of engrafted macrophages displayed following LPS challenge three ATACseq peaks (V–VII), that were significantly enhanced over host microglia (Fig. 6d). An induced peak located 54,402 bp upstream of the TSS displayed a TBX5 motif, whereas a second peak located 38,600 bp upstream of the Spp1 TSS displayed an AP-1 (Jun-related) motif (Fig. 6d). Collectively, these data establish that engrafted microglia respond differently from host microglia to a challenge and are hence functionally distinct.

**Engrafted macrophages in human brain differ from microglia**. Engrafted brain macrophages differ from host microglia by their gene expression (Fig. 3b). To confirm this finding for protein expression, we performed a histological analysis of brains of the BM chimeras generated by TBI and BU conditioning (Supplementary Fig. 2d). Engrafted cells and host microglia were identified by IBA-1 staining. Graft-derived cells were defined

according to eBFP expression conveyed by the lentiviral construct (Fig. 2k). In concordance with the transcriptome data (Fig. 3g), analysis for Tmem119 and P2ry12 expression revealed the absence of these markers from the graft (Fig. 7a, b), while eBFP+ cells displayed ApoE and MHC class II (Supplementary Fig. 10).

To finally extrapolate our finding to a human setting, we analyzed *post mortem* brains of patients who underwent HSC transplantation. Specifically, we took advantage of gender-mismatched grafts that allowed identification of the transplant by virtue of its Y-chromosomes through chromogenic in situ hybridization (CISH). Ramified IBA-1+ microglia-like cells harboring the Y chromosome could be readily identified juxtaposed to Y chromosome-negative host microglia in cortex, cerebellum, and hippocampus sections of the subject brains (Fig. 7c). Expression of the purinergic P2Y$_{12}$ receptor has been proposed earlier to serve as marker for human microglia[49,59]. Moreover, absence of the P2Y$_{12}$ receptor compromises microglial activation by nucleotides and could thus have functional implications[60]. As observed in the murine chimeras, the human P2Y$_{12}$ receptor expression was found to be restricted to host cells, but absent from Y chromosome-positive brain macrophages cells (Fig. 7d). Collectively, these data establish that in the CNS of patients who underwent HSC transplantation, graft-derived cells remained functionally distinct from host microglia and strengthen the conclusion that HSC-derived engrafted cells differ from host microglia.

## Discussion

Here we established that BM-derived brain macrophages that persistently seed the CNS of recipient organisms following irradiation or myeloablation remain distinct from host microglia with respect to their transcriptomes, chromatin accessibility landscapes, and response to challenge.

Following the engraftment of conditioned recipient mice, transplanted cells establish under the influence of the CNS microenvironment, a characteristic microglia transcriptome that distinguishes these cells from other tissue macrophages[47]. Thus, nine-tenth of their transcriptome is shared with host microglia, including expression of Fc receptor-like molecules (*Fcrls*) and Tgfb receptor (*Tgfbr*), as well as the MADS box transcription enhancer factor 2 (*Mef2c*). Moreover, residence in the CNS endowed engrafted cells with microglia characteristics, such as longevity, radio-resistance and ramified morphology. Engrafted cells however failed to adopt complete host microglia identity even after prolonged CNS residence. Corroborating and extending earlier reports[47,49,50], this included significantly reduced mRNA expression of the microglia markers Tmem119, SiglecH and P2yr12 and complete lack of the transcriptional repressors Sall1 and Sall3. Conversely, engrafted macrophages expressed genes absent from host microglia, including *Ccr2*, *Ifnar1*, *Msa4a7* and *ApoE*, and displayed Msr1 and Axl mRNAs, potentially related to the absence of Sall1[50]. Transcriptomes of engrafted macrophages showed considerable overlap with perivascular macrophages and indication of cell activation, such as an underrepresentation of a regulatory pathway driven by IL-10, as compared to host microglia. Comparative transcriptome analysis showed that our data are well in line with recent studies[54,55] that reported that also macrophages that engraft the brain of mice conditionally depleted of microglia due to a Csf1r deficiency retain a transcriptional identity distinct from host cells.

Analysis of open chromatin revealed that graft cells acquire a transposase-accessible profile that is similar to that of host cells and is enriched for a common set of motifs that are recognized by TFs known to be important for microglia development, such as PU.1 and MEF2c. However, host and graft cells also exhibit

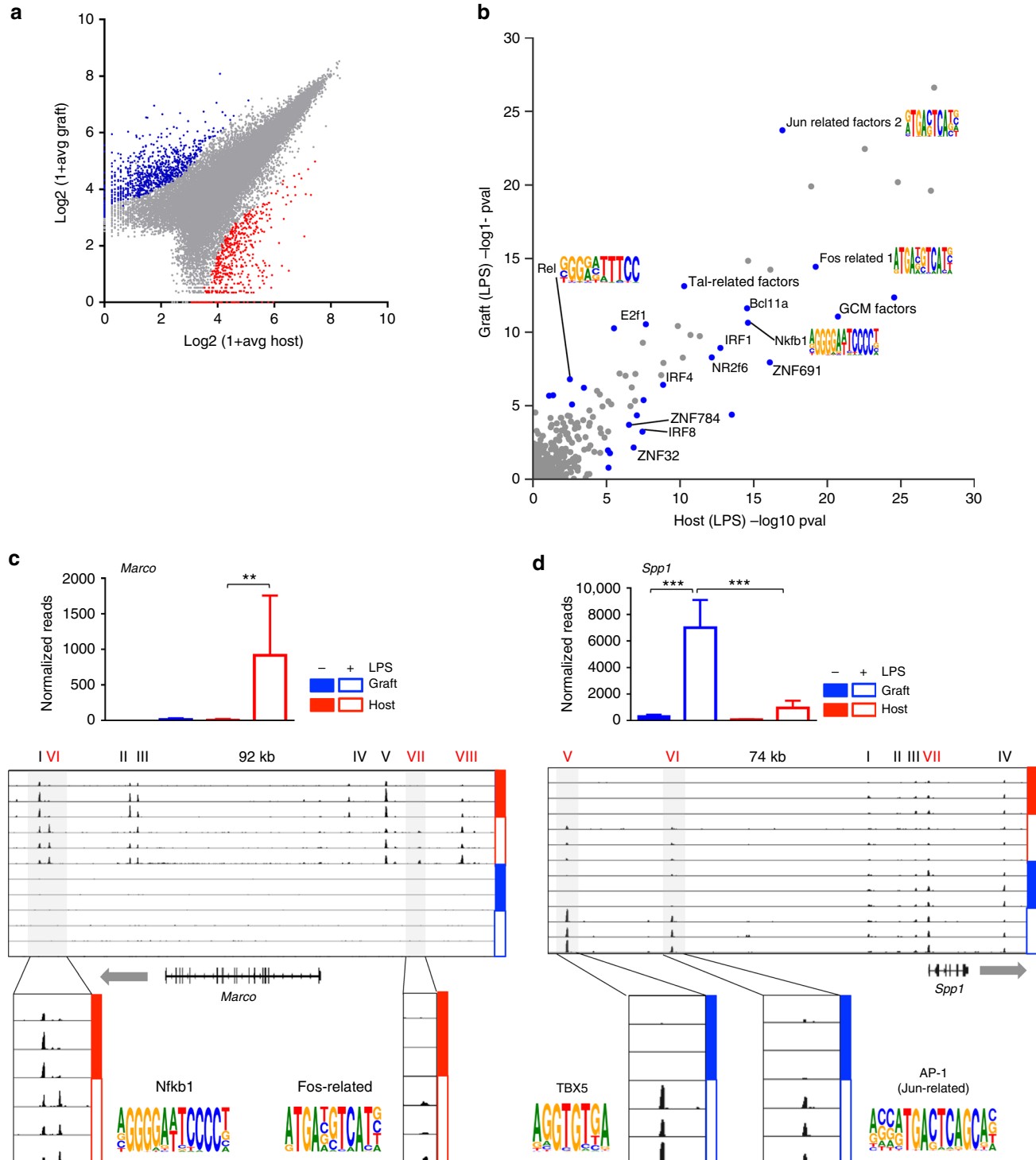

**Fig. 6** Comparative epigenome analysis of graft and host microglia post LPS challenge. **a** Analysis of all 46,485 detected ATAC peaks, from which 552 peaks and 841 peaks displayed >4-fold significant (*p*-value < 0.01) enrichment in host microglia and engrafted cells isolated from challenged mice, respectively. **b** Comparison of motif significance in activated/challenged host and grafted cells. *P*-values were calculated using TBA models trained on intergenic ATAC-seq peaks from host and grafted cells. Significant motifs that show a large difference (*p* < 10e−5, log-likelihood ratio > = 2) are indicated in blue points. Motif logos visualizing the position frequency matrix of NF-kappaB and AP-1 motifs are annotated. **c** Challenge induced alterations in *Marco* locus. Normalized sequence reads of *Marco* mRNA in engrafted cells and host microglia isolated from LPS challenged and unchallenged BM chimeras (top); normalized ATACseq profiles of *Marco* locus (bottom), with enlarged areas highlighting induced ATACseq peaks and predicted motifs. **d** Challenge induced alterations in *Spp1* locus. Normalized sequence reads of *Spp1* mRNA in engrafted cells and host microglia isolated from LPS challenged and unchallenged BM chimeras (top); normalized ATACseq profiles of *Spp1* locus (bottom), with enlarged areas highlighting induced ATACseq peaks and predicted motifs. Source data are provided as a Source Data file

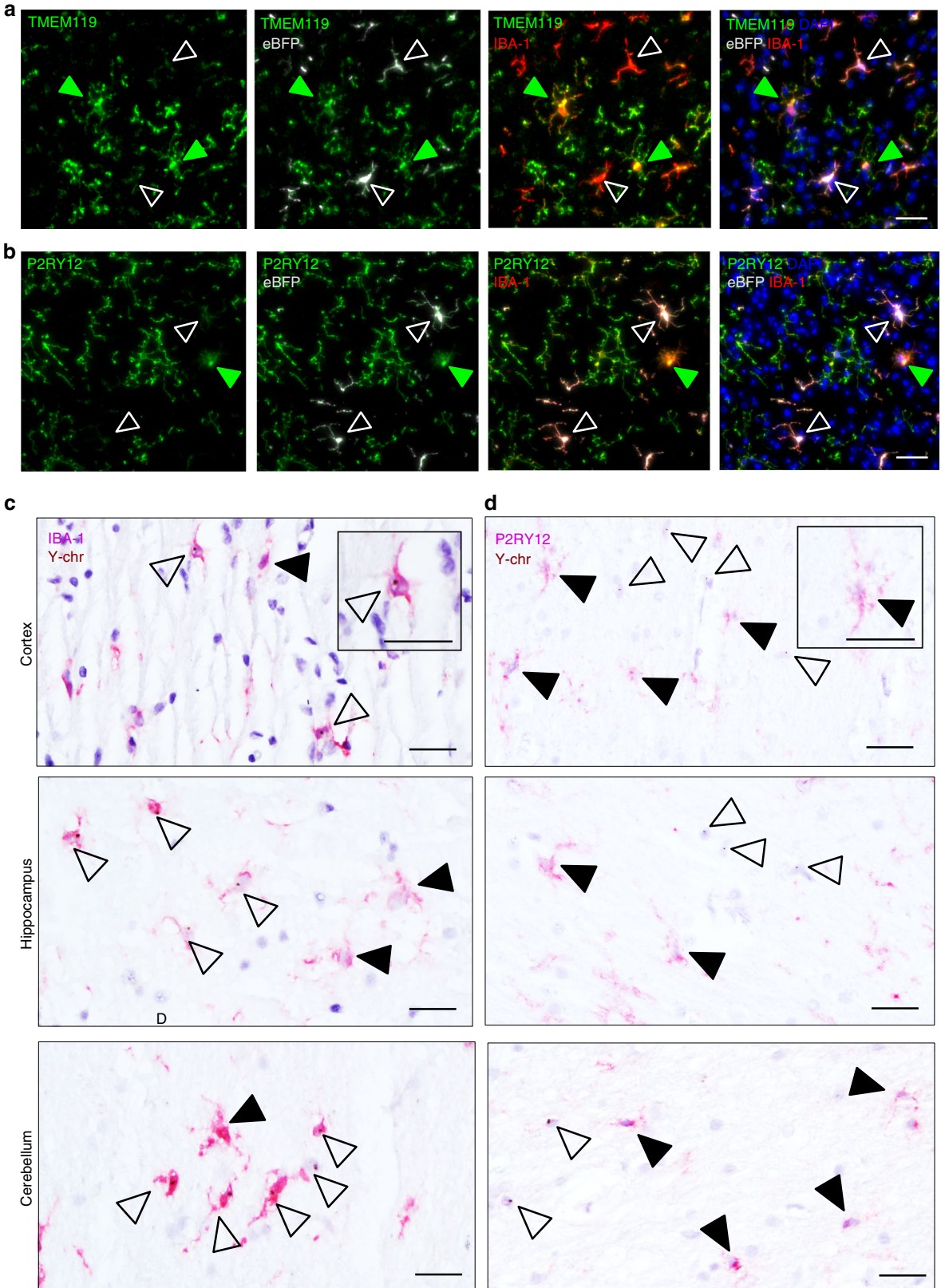

significant differences in open chromatin that are associated with distinct motif enrichment patterns and the observed differences in gene expression. These findings imply that the distinct developmental origins of host and graft cells determine the ability of the brain environment to fully activate the complement of

transcription factors required for microglia identity, most notably exemplified by lack of induction of *Sall1* in graft cells. The differences in chromatin landscapes under resting conditions are likely to contribute to the host and graft-specific responses to LPS challenge. This possibility is supported by the observation that

**Fig. 7** Comparative protein expression analysis of graft and host microglia in mouse and human chimeras. **a**, **b** Expression of host microglia-specific markers TMEM119 (**a**) and P2RY12 (**b**) in the cortex of mice that received lineage-negative BM carrying the lentiviral construct conferring eBFP expression. Host microglia (green) and donor cells (white) are indicated with arrowheads. IBA-1 immunohistochemistry for microglia (red). DAPI nuclear counterstain (blue). Scale bars, 30 μm. **c**, **d** Representative images of cortical, hippocampal, and cerebellar sections from female patients that received male donor BM grafts carrying the Y-chromosome (Y-chr). Host microglia are indicated by solid arrowheads and donor cells are shown by open arrowheads. Immunohistochemistry (red) of IBA-1 (**c**) and P2RY12 (**d**) combined with in situ hybridization of Y-chr (brown). Insets show a single grafted cell at higher magnification. Scale bars, 30 μm

alternative motifs for NFkB and AP-1 factors are enriched in open chromatin of host and graft cells. We interpret this finding to reflect the binding of NFkB and AP-1 dimers and heterodimers to different locations in the genome that are specified by the specified by host or graft-specific combinations of transcription factors, respectively.

The exact origin of the BM-derived engrafted cells in the chimeric organisms remains to be defined. In a classic study, Ajami and colleagues established that non-parenchymal brain macrophages that can persistently seed the host brain originate from non-monocytic BM-resident myeloid progenitors characterized by the absence of $CX_3CR1$ expression[26]. Likewise, other studies suggested that a transient wave of early hematopoietic progenitors infiltrates the host CNS during transplantation and following local proliferation, establishing the graft[31]. This notion is supported by the results of our 'Microfetti' and barcoding approaches that establish that engrafted macrophages undergo clonal proliferation and thereby likely progressively outcompete irradiation or busulfan-damaged host microglia. Moreover, the conclusion that engrafted cells arose from cells that do not contribute to long-term hematopoiesis in the chimeras is also in line with the prominent detection of private clones in this population, which are not shared with the other hematopoietic compartments.

Future studies could aim to identify cells that upon engraftment will give rise to closer mimics of host microglia, including for instance expression of Sall1. This could include cells linked with the unique developmental YS origin of microglia[54], or otherwise manipulated cells such as microglia-like cells derived from induced-pluripotent-stem-cell (iPS)-derived primitive macrophages[61]. Furthermore, in the context of gene therapy, viral vectors could be used to express transgenes to engineer the engrafted cells to boost engraftment and modulate their function. Of note however, under certain pathological conditions the distinct engrafted BM-derived macrophages we report might also be advantageous as compared to host microglia. BM-derived cells could for instance be superior to YS-derived microglia in the handling of the debris burden associated with senescence[62] or amyloid plaques that arise during Alzheimer's disease[63]. Elucidation of such scenarios should profit from the molecular definition of the engrafted cells and host microglia like the one provided in this study. However, in vivo functions of microglia remain poorly understood and future dedicated experimentation will be required to compare the performance of engrafted macrophages and host microglia in different disease models during aging and specific challenges.

Recent studies revealed a signature of disease associated microglia, termed 'DAM'[52] or 'MGnD'[64], that is induced by various brain-intrinsic changes in the absence of massive peripheral infiltrates, though not following peripheral LPS challenge[58,65], and can occur on acute to chronic time scales[58]. Of note, engrafted brain macrophages displayed robust constitutive expression of some of theses DAM genes, such as *ApoE* and *Axl*. Likewise, as opposed to microglia[58], engrafted BM-derived brain macrophages responded to the LPS challenge with the induction of DAM/ MGnD hallmarks, such as the CD44 ligand Spp1/ Osteopontin. The latter might have implications when considering brain macrophage contributions to CNS pathologies. Moreover, our data support the notion that expression of genes included in the DAM/MGnD signature are in microglia under stringent control, potentially including repression by Sall1.

Results obtained from fate mapping models currently suggest that at least in the brain of unchallenged C57BL/6 mice kept in specific-pathogen-free (SPF) facilities, parenchymal macrophages are exclusively comprised of YS-derived microglia. Further experimentation will however be required to assess how absolute this exclusion of HSC-derived macrophages is, in particular following challenges. Moreover, it remains currently unclear to what extend HSC-derived macrophages might be able to seed the human brain, f.i. during extended aging, and could hence impact brain pathologies.

Tissue macrophages, such as Kupffer cells (KC) and alveolar macrophages (AM), have been reported to be faithfully replaced by BM-derived cells in irradiation chimeras and other small animal models involving deficiencies of the resident compartment[6–9]. While these studies were restricted to transcriptome comparison and hence might have missed epigenetic differences between graft and host cells, the inability of HSC-derived cells to achieve full host cell identity might be unique to microglia and related to features particular to these cells. Specifically, among adult tissue macrophages, only microglia derive from primitive YS macrophages and this origin could define cell identity. In contrast, generation of both KC and AM involves a monocytic intermediate, and their re-generation might hence be attainable by the closer related HSC-derived cells that can also give rise to monocytes. Alternatively, establishment of the 'bona fide' microglia signature might require 'physiological' microglia development in the developing CNS that is associated with profound transient activation of this brain macrophage compartment[12,13,57,66].

Collectively, the demonstration that engrafted cells and host microglia remain distinct sheds light on the molecular and functional heterogeneity of parenchymal brain macrophages. Moreover, when extrapolated to the human setting, our findings could have major implications for patients treated by HSC gene therapy to ameliorate lysosomal storage disorders, microgliopathies or general monogenic immuno-deficiencies.

## Methods

**Mice.** For generation of BM chimeras, wild type C57BL/6 J mice (Harlan) were used as recipients, Cx3cr1^GFP or CX₃CR1^Cre:R26-RFP^fl/fl mice[32,67] were used as donors. Recipient mice were lethally irradiated with a single dose of 950 cGy using an XRAD 320 machine (Precision X-Ray (PXI)) and reconstituted the next day by i.v. injection of $5 \times 10^6$ donor BM cells per mouse. All animals bred at the Weizmann animal facility were maintained under specific pathogen-free conditions and handled according to protocols approved by the Weizmann Institute Animal Care Committee as per international guidelines. Female C57BL/6 J wild type recipient mice (Charles River) received full body irradiation (7 Gy) in the RS 2000 Biological Research Irradiator at 8 weeks of age. After 24 h, $5 \times 10^6$ donor BM cells isolated from Cx3cr1^CreER/+:R26R^Confetti/+ 'Microfetti' mice[34] were injected into the tail vein of recipients. Recipient mice received 500 µl Cotrim K-ratiopharm® antibiotics in 250 ml drinking water for 2 weeks after BM reconstitution. Tamoxifen (Sigma) dissolved in corn oil (Sigma) was subcutaneously applied in a single dose of 10 mg distributed along the flanks. Mice were maintained in specific-pathogen-free facility with chow and water provided ad libitum. Animal experiments performed in

Freiburg were approved by the Regional Council of Freiburg, Germany. All experimenters were blinded during data acquisition and analysis.

**Barcoding experiment.** Female 8-weeks-old C57Bl6J wild-type recipient mice (CD45.2) were conditioned with either total body irradiation (9.5 Gy) or 125 mg busulfan per kg bodyweight (in 5 doses of 25 mg each)[38] prior to transplantation of $5 \times 10^5$ lineage negative cells from male CD45.1-mice (8 weeks of age), transduced with lentiviral BC32-eBFP vectors[35]. Intermediate peripheral blood samples were taken every 4–6 weeks to monitor chimerism and marking efficiency via flow cytometry. Six months posttransplantation, the mice were sacrificed and peripheral blood, bone marrow, spleen and the brain were taken for clonal analyses. Mice were maintained in specific-pathogen-free facility with chow and water provided ad libitum. Animal experiments performed in Hamburg were approved by the local authorities (Behoerde fuer Gesundheit und Verbraucherschutz-Veterinaerwesen/ Lebensmittelsicherheit).

**Lipopolysaccharide challenge.** For lipopolysaccharide (LPS) treatment, mice were injected intra-peritoneally (i.p.) with a single dose of LPS (2.5 mg/kg, E. coli 0111: B4; Sigma) and sacrificed 12 h post-injection.

**Microglia isolation.** To isolate microglia and BM-derived parenchymal CNS macrophages, BM chimeras were perfused using ice-cold phosphate buffered saline (PBS) and brains were harvested. Brains were dissected, homogenized by pipetting and incubated for 20 min at 37 °C in a 1 ml HBSS solution containing 2% BSA, 1 mg/ml Collagenase D (Sigma) and 1 mg/ml DNase1 (Sigma). The homogenate was then filtered through a 100 μm mesh and centrifuged at 2200 RPM, at 4 °C, for 5 min. For the enrichment of microglia and BM-derived cells, the pellet was re-suspended with a 40% percoll solution (Sigma) and centrifuged at 2200 RPM, room temperature for 15 min. The cell pellet was next subjected to antibody labeling and flow-cytometry analysis.

**Flow cytometry and cell sorting.** Cells were stained with primary antibodies against CD45.1 (A20), CD45.2 (104), CD11b (M1/70), Ly6C (AL-21), and LY6G (1A8)—all from Biolegend, San Diego, CA, USA. After incubation with the Abs at 4 °C for 15 min, cells were washed and sorted using a FACSAria (BD, Erembodegem, Belgium) flow cytometer. Data were acquired with FACSdiva software (Becton Dickinson). Post-acquisition analysis was performed using FlowJo software (Tree Star, FlowJo LLC; Ashland, Oregon).

**RNA-seq analysis.** RNA-seq of populations was performed as described previously[10]. In brief, $10^3$–$10^5$ cells from each population were sorted into 50 μl of lysis/binding buffer (Life Technologies) and stored at 80 °C. mRNA was captured with Dynabeads oligo(dT) (Life Technologies) according to manufacturer's guidelines. We used a derivation of MARS-seq[41] to prepare libraries for RNA-seq. Briefly, RNA was reversed transcribed with MARS-seq barcoded RT primer in a 10 μl volume with the Affinity Script kit (Agilent). Reverse transcription was analyzed by qRT-PCR and samples with a similar CT were pooled (up to eight samples per pool). Each pool was treated with Exonuclease I (NEB) for 30 min at 37 °C and subsequently cleaned by 1.2× volumes of SPRI beads (Beckman Coulter). Subsequently, the cDNA was converted to double-stranded DNA with a second strand synthesis kit (NEB) in a 20 mL reaction, incubating for 2 h at 16 °C. The product was purified with 1.4× volumes of SPRI beads, eluted in 8 μl and in vitro transcribed (with the beads) at 37 °C overnight for linear amplification using the T7 High Yield RNA polymerase IVT kit (NEB). Following IVT, the DNA template was removed with Turbo DNase I (Ambion) 15 min at 37 °C and the amplified RNA (aRNA) purified with 1.2 volumes of SPRI beads. The aRNA was fragmented by incubating 3 min at 70 °C in $Zn^{2+}$ RNA fragmentation reagents (Ambion) and purified with 2× volumes of SPRI beads. The aRNA was ligated to the MARS-seq ligation adapter with T4 RNA Ligase I (NEB). The reaction was incubated at 22 °C for 2 h. After 1.5× SPRI cleanup, the ligated product was reverse transcribed using Affinity Script RT enzyme (Agilent) and a primer complementary to the ligated adapter. The reaction was incubated for 2 min at 42 °C, 45 min at 50 °C, and 5 min at 85 °C. The cDNA was purified with 1.5× volumes of SPRI beads. The library was completed and amplified through a nested PCR reaction with 0.5 mM of P5_Rd1 and P7_Rd2 primers and PCR ready mix (Kappa Biosystems). The amplified pooled library was purified with 0.7× volumes of SPRI beads to remove primer leftovers. Library concentration was measured with a Qubit fluorometer (Life Technologies) and mean molecule size was determined with a 2200 TapeStation instrument. RNA-seq libraries were sequenced using the Illumina NextSeq 500. Raw reads were mapped to the genome (NCBI37/mm9) using hisat (version 0.1.6). Only reads with unique mapping were considered for further analysis. Gene expression levels were calculated and normalized using the HOMER software package (analyzeRepeats.pl rna mm9 -d<tagDir>-count exons -condenseGenes -strand + -raw)[68]. Differential expressed genes were selected using a 2-fold change cutoff between at least two populations and adjusted p value for multiple gene testing > 0.05. Gene expression matrix was clustered using k-means algorithm (MATLAB function kmeans) with correlation as the distance metric. The value of k was chosen by assessing the average silhouette (MATLAB function silhouette) (3) for a range of possible values (4–15).

**ATAC-seq analysis.** 20,000–50,000 cells were used for ATAC-seq[42] applying described changes[69]. Briefly, nuclei were obtained by lysing the cells with cold lysis buffer (10 mM Tris-HCl pH 7.4, 10 mM NaCl, 3 mM MgCl₂, 0.1% Igepal CA-630) and nuclei were pelleted by centrifugation for 20 min at 500 × g, 4 °C using a swing rotor. Supernatant was discarded and nuclei were re-suspended in 25 μl reaction buffer containing 2 μl of Tn5 transposase and 12.5 μl of TD (Nextera Sample preparation kit from Illumina). The reaction was incubated at 37 °C for 1 h. DNA was released from chromatin by adding 5 μl of cleanup buffer (900 mM NaCl, 300 mM EDTA, 1.1% SDS, 4.4 mg/ml Proteinase K (NEB)) followed by an incubation for 30 min at 40 °C. Tagmentated DNA was isolated using 2× volumes of SPRI beads and eluted in 21 μl. For library amplification, two sequential PCRs (nine cycles, followed by an additional six cycles) were performed in order to enrich small tagmentated DNA fragments. We used the indexing primers as described by Buenrostro et al. and KAPA HiFi HotStart ready mix. After the first PCR, the libraries were size-selected using double SPRI bead selection (0.65× followed by 1.8×). Then the second PCR was performed with the same conditions in order to obtain the final library. DNA concentration was measured with a Qubit fluorometer (Life Technologies) and library sizes were determined using TapeStation (Agilent Technologies). Libraries where sequenced on the Illumina NextSeq 500 obtaining an average of 20 million reads per sample. Putative open chromatin regions (peaks) were called using HOMER[68] (using parameters compatible with IDR analysis: –L 0 –C 0 –fdr 0.9). The irreproducible discovery rate (IDR) was computed for each peak using the Homer peak score for each replicate experiment (https://github.com/nboley/idr); peaks with IDR > 0.05 were filtered away. Normalization and differential expression analysis was done using the DESeq2 R-package. Intergenic peaks annotated by Homer were used to train TBA models for each cell type using default parameters (https://github.com/jenhantao/tba). The significance of each motif in each cell type was assigned by comparing the predictive performance of the trained TBA model and a perturbed model that cannot recognize one motif using the chi-squared test.

**Motif screen with transcription factor binding analysis.** We used Transcription factor Binding Analysis (TBA) models to identify enriched motifs in open chromatin in comparison to a set of randomly selected genomic loci (matched for GC content). For each of the sequences in the combined set of the ATAC-seq peaks and background loci, TBA calculates the highest motif score for each of the motifs (in either orientation) included in the model. The sequences of open chromatin regions and background loci as well as the corresponding motif scores are used to train the TBA model to distinguish open chromatin from background loci (using five-fold cross validation). A TBA model scores the probability of observing open chromatin given a genomic sequence by computing a weighted sum over all the motif scores computed for that sequence. The weight for each motif is learned by iteratively modifying the weights (starting from random values) until the model's ability to differentiate open chromatin from background sequences no longer improves. The significance of a given motif was assigned by comparing the predictive performance of a trained TBA model and a perturbed model that cannot recognize that motif using the likelihood ratio test. We trained TBA models for each cell type, using version 1.0 of TBA and default parameters (source code and executable files are available at: https://github.com/jenhantao/tba). For a complete description see ref. [56].

**Histology.** Mice were transcardially perfused with PBS followed by 4% paraformaldehyde in PBS. Mouse brains were post-fixed for 6 h or overnight at 4 °C and processed for frozen sectioning as previously detailed[34]. Free floating 50-μm cryosections and 14-μm cryosections on slides were prepared from Microfetti and lineage negative barcoded BM chimeras, respectively. Tissues were permeabilized in blocking solution (0.1% Triton-X-100, 5% bovine albumin, and PBS) for 2 h at room temperature and incubated overnight at 4 °C with primary antibodies: 1:500 rabbit anti-Iba-1 (Wako), 1:200 goat anti-Iba-1 (Novus), 1:1000 chicken anti-GFP against eBFP (Abcam), 1:1000 rabbit anti-TMEM119 (Abcam), 1:500 rabbit anti-P2RY12 (Ana Spec), 1:200 goat anti-APOE (Millipore), and 1:100 mouse anti-MHC Class II (Abcam). Antigen retrieval was performed prior to APOE staining for 40 min at 92 °C in pH 9 citrate buffer. Corresponding secondary antibodies conjugated to Alexa Fluor 488, Alexa Fluor 568, or Alexa Fluor 647 (Life Technologies) were applied at 1:1000 with nuclear counterstain by 4′,6-diamidino-2-phenylindole (DAPI, Sigma) at 1:5000 for 2 h at room temperature. Sections were mounted in ProLong® Diamond Antifade Mountant (Life Technologies).

**Image acquisition and analysis.** Brain images of Microfetti and lineage negative BM chimeras were acquired on the Keyence BZ-9000 inverted fluorescence microscope using a 20X/0.75 NA objective lens. Images were processed and analyzed using ImageJ (NIH). Intercellular distances were determined by Calculate-NearestNeighbor from Kota Miura (https://doi.org/10.5281/zenodo.1323726). Cell morphological analyses were performed with ImageJ plugins Simple Neurite Tracer[70] and Sholl Analysis[71] using default settings.

**Human gender-mismatched stem cell transplantation patients.** Experiments on human tissue samples were performed according to the Declaration of Helsinki. Ethical approval was obtained from the local Research Ethics

Committee of the University Medical Center of Freiburg (ref. no. 10008/09). The analyzed brain tissue samples of female patients, who underwent gender-mismatched peripheral blood stem cell transplantation (PBSCT), were derived from the brain autopsy case archive of the Institute of Neuropathology, University of Freiburg. Patient 1 was diagnosed with myelo-dysplastic syndrome (MDS) 12 years before death and received a first sex-matched PBSCT 6 years later. Graft failure developed a year later. The patient's MDS transformed into acute myeloid leukemia (AML) 17 months prior to death. Patient 1 then received a second PBSCT from a human leukocyte antigen (HLA)-identical male donor. Transplantation was initiated after a myelo-ablative therapy regimen comprising Thiotepa, Fludarabine, and Treosulfan. Complications after transplantation included mucositis and cytomegalovirus reactivation. Complete chimerism was confirmed in two follow-up examinations. Patient 1 died at the age of 66 years 453 days after sex-mismatched PBSCT due to pneumonia and pulmonary leukostasis caused by AML infiltration. Patient 2 was diagnosed with multiple myeloma 4.5 years before death and first treated with several chemotherapy regimens. She received PBSCT from a HLA-identical male donor 1.5 years after diagnosis. Myeloablation was performed with Thiotepa, Fludarabine, and Busulfan. After transplantation, Patient 2 developed graft-vs.-host disease of the skin and gut. Follow-up examinations confirmed complete chimerism. Patient 2 died at the age of 58 years 1018 days after sex-mismatched PBSCT due to pneumonia causing respiratory failure.

**Combined immunohistochemistry and in situ hybridization**. After the death of the two patients, brains were transferred into 4% paraformaldehyde (PFA) within 48 h and fixed for at least 3 weeks. After fixation, representative tissues from several brain regions including the temporal cortex, hippocampus, and cerebellum were dissected and embedded in paraffin. Routine neuropathological examination revealed minimal signs of neurodegeneration in Patient 1, according to Braak stage I-II[72]. Patient 2 showed no signs of brain pathology. Combined immunohistochemistry (IHC) and chromogenic in situ hybridization (CISH) were performed on 10 μm thick sections. Sections were deparaffinized and heated at 92 °C, at pH 6 for 40 min for antigen retrieval. For IHC, the sections were incubated with the respective primary antibodies for 30 min at room temperature: 1:1000 rabbit anti-IBA-1 (Abcam, clone EPR 16588), 1:1500 rabbit anti-P2RY12 (Sigma-Aldrich, polyclonal). Secondary goat anti-rabbit antibodies (Southern Biotech) were applied at 1:200 for 1 h at room temperature. Liquid Permanent Red Substrate-Chromogen (Agilent Dako) was used for visualization of the antigen. After 5 min post-fixation in 4% PFA and rinse in deionized water, CISH was performed on the sections using the ZytoDot CISH Implementation Kit (ZytoVision) according to manufacturer's instructions with the following modifications. After incubation in ethylene diamine tetra-acetic acid (EDTA) at 95 °C for 15 min, sections were treated with pepsin solution at 37 °C for 6 min. After stepwise dehydration, 12 μl of ZytoDot CEN Yq12 digoxigenin-linked probe (ZytoVision) was added to each brain section to bind to the Yq12 region of the human Y-chromosome for at least 20 h at 37 °C. Following the washing and blocking steps, the sections were incubated in mouse anti-digoxigenin antibody solution, treated with horseradish peroxidase-conjugated anti-mouse antibody at 37 °C for 30 min, and bound to 3,3′-diaminobenzidine at 37 °C for 45 min. Nuclei were counterstained with hematoxylin. Images were acquired using MikroCam II with a UPlan FLN 40×/0.75 NA objective on a BX40 microscope (Olympus).

**Barcode analyses**. DNA was extracted from peripheral blood, bone marrow, spleen and sorted microglia and CD45$^{high}$ cells were used for barcode amplification, multiplexing, as well as the bioinformatic processing[36]. Unique barcodes with more than 100 reads per sample were taken into further analyses. Venn diagrams were produced by an in-house R-script using the "VennDiagram" package.

**Digital droplet PCR**. To determine the chimerism in the sorted microglia, digital droplet PCR was performed. In a duplex reaction, a Y-chromosome-specific fragment (and a control amplicon (located in the erythropoietin receptor) were simultaneously amplified and analyzed using the QX200 system (BioRad).

**Statistical analysis**. Mean data are shown. Mann–Whitney test, D'Agostino–Pearson test, two-tailed unpaired $t$ test, and Welch's $t$ test were performed in GraphPad Prism7. Statistical significance was taken at $p < 0.05$.

## Data availability

The accession codes for the RNA-seq and ATACseq datasets reported in this paper can be found at GEO: GSE122769. Additional data that support the findings of this study are available as Source Data for Fig. 3b–d, h, Fig. 4d–f, Fig. 5a–d, Fig. 6a–d, Supplementary Figures 7b, 8a, and 10b. Other data are available from the corresponding author upon request.

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

## Acknowledgements

The Jung laboratory was supported by the Israeli Science Foundation (887/11), the European Research Council (Adv ERC 340345), the Deutsche Forschungsgemeinschaft (DFG) (CRC/TRR167 'NeuroMac'), the U.S.-Israel Binational Science Foundation (BSF) and a collaborative network grant of the International Progressive MS Alliance (PMSA). M.P. is supported by the BMBF-funded competence network of multiple sclerosis (KKNMS), the Sobek Foundation, the Ernst-Jung Foundation, the DFG (SFB 992, SFB1160, SFB/TRR167, Reinhart-Koselleck-Grant) and the Ministry of Science, Research and Arts, Baden-Wuerttemberg (Sonderlinie "Neuroinflammation"). T.L.T. was supported by the German Research Foundation (DFG, TA1029/1-1) and Ministry of Science, Research and the Arts of Baden-Württemberg (7532.21/2.1.6). The Glass laboratory was support by NIH grants NS096170, DK091183 and GM085764. The authors thank G. Friedlander for help with bioinformatics, T. Sonntag, E. Orthey and the UKE FACS Core Facility from the University Medical Center Hamburg-Eppendorf, as well as J. Dautzenberg and E. Barleon from the University of Freiburg for excellent technical assistance.

## Author contributions

A.S., T.L.T., A.V., J.-S.K., M.G., K.C., and P.S. performed experiments; J.G., J.T., E.D., and L.T. performed bioinformatics analysis; A.A.-F. and L.C.-M. provided technical help; C. K.G., K.C., and M.P. supervised and advised; S.J. wrote the manuscript, secured funding, and supervised.

## Additional information

**Competing interests:** The authors declare no competing interests.

