## [Peer Review File · Nature Communications]

Reviewers' comments:

Reviewer #1 (Microglia, brain-immune)(Remarks to the Author):

In the research paper by Shemer et al., the authors show that in mice, engrafted bone marrow- and HSC-derived cells acquire a microglia-like phenotype in the brain, but remain distinct from yolk sac-derived host microglia even after long-term engraftment. Transcriptomic identity and epigenetic characteristics of bone marrow-derived macrophages are also maintained after endotoxin challenge despite the similarity in their inflammatory responses.

This paper is an important contribution for the field of microglia biology. Beyond some repetitive observations also presented in other recent works, the authors convincingly show that efficiently engrafted donor cells not only adopt the phenotype of resident microglia, but suggest that most grafted cells originate from precursors that seed the host CNS early after engraftment and maintained independent from ongoing hematopoiesis. It is also of importance that grafted cells adopt epigenetic characteristics similar to microglia, but they remain distinct from host cells even after prolonged CNS residence, including their responses to inflammatory challenge. In addition, the authors show that in patients who had received gender mismatched grafts, purinergic P2Y₁₂ expression is restricted to host cells, but absent from Y chromosome-positive graft-derived „microglia“.

Major issues:

- As also seen in other recent works, the authors convincingly show that engrafted cells adopt microglia characteristics, such as relative radio-resistance, longevity and ramified morphology. One novel aspect of the present work comes from the tandem engraftment study, which allows the assessment of the long-term survival of blood-borne macrophages. However, it seems that the second engraftment leads to far better survival of the second graft (RFP+) than those cells derived from the first BM transplantation (GFP+). Is it because engrafted cells are in fact not very radioresistant, or because the capacity of the cells to engraft the brain increases due to repeated irradiation challenges (BBB injury, etc)?

- The genetic barcoding experiment suggests that engrafted donor cells not only adopt the phenotype and distribution of resident microglia, but most grafted cells originate from precursors that seed the host CNS early after engraftment and maintained independent from ongoing hematopoiesis. There are two issues here that require explanation. First, some disturbance of the microglial niche is required for macrophages to efficiently engraft the brain.

Did the authors find any indication to injury or death of resident microglia soon (1-2 weeks) after the transfer of HSC/HPCs and busulfan treatment or would early-engrafting macrophages appear as additional cells in the brain in spite that the microglial niche is occupied by otherwise healthy, resident microglia? Possible injury to microglia is also mentioned in the discussion, however, this is not explored in any way. This issue is important since after parabiosis (when the microglial niche is not disturbed) there seems to be very little contribution of blood-borne macrophages to the brain macrophage pool. Second, as the authors also propose, engrafted macrophages are likely to undergo clonal proliferation in the brain if they originate from precursors that are recruited early on. However, this is not demonstrated in this study. Has BrdU labelling or any similar approach been used to prove that proliferation of exogenous macrophages in fact takes place in the brain parenchyma?

- The authors provide very little information about the actual location of engrafted cells compared to resident microglia in the context of blood vessels and brain ventricles. They mention concerning the Microfetti study that „Seamless integration of engrafted cells into the endogenous microglial network was reflected by their Iba-1 expression, similar morphology and intercellular distances as compared

host microglia", but do not show that these cells display similar distribution to resident microglia or that intercellular distances are similar. Instead, on the images presented the cells appear in clusters in most brain areas and not too many of them can be seen in the cortex and the hippocampus compared to the numbers in the olfactory bulb and cerebellum, where the turnover of microglia is reportedly faster. This would argue against the real parenchymal integration of these cells. Previous BM transplantation studies have demonstrated the preferential proximity of blood-borne macrophages to ventricles and blood vessels, but it remained unclear whether changes in the brain microenvironment by irradiation itself played a role in later distribution of engrafting cells. Has any comparison between busulfan treatment and irradiation been performed in this regard (as for example suggested by Wilkinson et al., Mol Ther. 2013)?

- Many studies have used simple or sophisticated anatomical tools to discriminate different activation stages of resident brain microglia in response to inflammatory challenges or disease. Have the authors tried to perform Sholl analysis or similar automated analysis to compare the branching characteristics and general morphology of microglia and engrafted macrophages based on Iba1 immunofluorescence after several months of engraftment? Such analysis would strengthen the statement regarding that engrafting macrophages adopt microglia characteristics. These data would be also very valuable for studies not able to discriminate resident microglia from exogenous macrophages based on genetic or fluorescent markers.

- Similarly to recent papers by others, the present study shows that while host microglia and engrafted macrophages display significant transcriptome overlap (partially due to the impact of the brain microenvironment on the cells), both microglia and macrophages maintain their transcriptomic identity. However, it is also proposed in the paper that grafted CNS macrophages displayed an activation signature. From a future therapeutic point of view this issue is important as this data suggest that even several months after transplantation engrafting macrophages would not fully adapt to the brain microenvironment. The reasons for this is currently unclear, but the authors should further elaborate on this based on their transcriptomic data sets and discuss this issue in the manuscript.

Further points:

- It would be helpful to also show quantitative data about the number of CD45^{high} monocyte populations in the brain from graft1, graft2 and resident cells for the proper interpretation of the data presented in Fig.1.
- Fig.1D. Graph supposedly shows Ly6C/G-negative cells as stated in the legend, unlike seen in the figure itself
- Fig.1E would be more informative with the addition of a third channel showing all microglia visualized by immunofluorescence (Tmem119, P2y12, etc).
- In the genetic barcoding experiment the authors used busulfan instead of irradiation, hence no major BBB injury is expected to take place in these studies. However, it would be helpful if the authors would demonstrate this in the paper.
- The level of significance between groups in the case of graphs presented in Fig.5C is unclear.
- The quality of P2RY12 staining presented in Fig.7D is suboptimal, especially compared to the importance of the statement that resident P2Y12-positive microglia lack the Y-chromosome.
- The text „Mean data are shown.“ appears in duplicate in the Methods. Welch Two (tailed?) t-test also needs attention.

Reviewer #2 (Immune-neuro crosstalk, NK)(Remarks to the Author):

This is an interesting work, demonstrating that BM-derived cells that engraft the brain after irradiation or myeloablation are distinct from the resident microglia in their transcriptome, epigenome and response to immunological stimulus. The authors conclude that the engrafting cells, although take over the niche, do not acquire full microglia phenotype. The consequence for brain function of such engraftment is not demonstrated and it is therefore unclear whether the function of these new cells is different or similar to yolk-sac derived microglia.

While the work is interesting although mostly unsurprising, the major weakness is its novelty. Works over the last few months have demonstrated, using a more elegant than irradiation method (namely, deletion of M-CSF from microglia and thus rendering them incapable of proliferation and refilling the niche), identical results upon BM-derived cell engraftment or upon injection of different progenitors. These two works, although cited in the paper, are only briefly mentioned. The authors are encouraged to discuss these two works in more details and preferably compare their results to those in previous publication (including transcriptomic analyses of the engrafted cells). Having more functional outcomes (behavioral aspects, electrophysiology, astrocyte physiology, etc) as a result of microglia replacement by cells of another origin, would make this work more exciting and innovative.

Reviewer #3 (Epigenetic)(Remarks to the Author):

I have read with interest the elegant work by Steffen Jung's group. Their conclusion that engrafted parenchymal brain macrophages differ from host microglia in transcriptome, epigenome and LPS responsiveness is convincing, and its significance is enhanced by supporting findings in human recipients of hemopoietic stem cell transplants. Overall, the epigenomic and transcriptomic data, on which I was asked to focus, are also robust and support the general conclusions of the paper. However, these analyses could be improved by integrating the RNASeq/ATACseq combination with targeted assessments of post-translational histone modifications and Chip-based approaches to specifically characterize regions/loci of interest. Indeed, as the authors acknowledge, "ATACseq does not discriminate between bound transcriptional activators and repressors". Therefore, some differentially expressed loci did not show epigenetic differences between engrafted parenchymal brain macrophages and host microglia. For some of those loci (and for the most compelling candidate loci that did show epigenetic differences), further characterization of histone marks and identification of transcription factors docking at sites of interest would highlight the regulatory landscape and would therefore be preferable to the purely bioinformatic approach currently adopted. Such studies would decisively deepen our understanding of the extent to which engrafted macrophages and host microglia diverge in regulatory terms.

Point by Point reply

Reviewer #1

In the research paper by Shemer et al., the authors show that in mice, engrafted bone marrow- and HSC-derived cells acquire a microglia-like phenotype in the brain, but remain distinct from yolk sac-derived host microglia even after long-term engraftment. Transcriptomic identity and epigenetic characteristics of bone marrow-derived macrophages are also maintained after endotoxin challenge despite the similarity in their inflammatory responses.

This paper is an important contribution for the field of microglia biology. Beyond some repetitive observations also presented in other recent works, the authors convincingly show that efficiently engrafted donor cells not only adopt the phenotype of resident microglia, but suggest that most grafted cells originate from precursors that seed the host CNS early after engraftment and maintained independent from ongoing hematopoiesis. It is also of importance that grafted cells adopt epigenetic characteristics similar to microglia, but they remain distinct from host cells even after prolonged CNS residence, including their responses to inflammatory challenge. In addition, the authors show that in patients who had received gender mismatched grafts, purinergic P2Y₁₂ expression is restricted to host cells, but absent from Y chromosome-positive graft-derived „microglia”.

We thank reviewer for acknowledging the importance of our work.

Major issues:

- As also seen in other recent works, the authors convincingly show that engrafted cells adopt microglia characteristics, such as relative radio-resistance, longevity and ramified morphology. One novel aspect of the present work comes from the tandem engraftment study, which allows the assessment of the long-term survival of blood-borne macrophages. However, it seems that the second engraftment leads to far better survival of the second graft (RFP+) than those cells derived from the first BM transplantation (GFP+). Is it because engrafted cells are in fact not very radioresistant, or because the capacity of the cells to engraft the brain increases due to repeated irradiation challenges (BBB injury, etc)?

This is an interesting point. However we can here only speculate. Following the 2nd irradiation peripheral cells of the first graft, incl. HSC, are depleted. Some brain macrophages derived from the first graft persist, which indicates relative radio-resistance. As compared to the 2nd graft these cells however are now themselves irradiated and hence probably have a disadvantage to compete with the new un-irradiated graft.

- The genetic barcoding experiment suggests that engrafted donor cells not only adopt the phenotype and distribution of resident microglia, but most grafted cells originate from precursors that seed the host CNS early after engraftment and maintained independent from ongoing hematopoiesis. There are two issues here that require explanation.

First, some disturbance of the microglial niche is required for macrophages to efficiently engraft the brain. Did the authors find any indication to injury or death of resident microglia soon (1-2 weeks) after the transfer of HSC/HPCs and busulfan treatment or would early-engrafting macrophages appear as additional cells in the brain in spite that the microglial niche is occupied by otherwise healthy, resident microglia? Possible injury to microglia is also mentioned in the discussion, however, this is not explored in any way. This issue is important since after parabiosis (when the microglial niche is not disturbed) there seems to be very little contribution of blood-borne macrophages to the brain macrophage pool.

Irradiation and busulfan are well established to cause genotoxic damage and cell death, and condition recipients for engraftment. These treatments hence likely provide the disturbance of the microglia niche that is required for seeding by grafted cells.

Indeed, parabiosis experiments seem to indicate that unless the niche is damaged there is no engraftment by HSC-derived cells. However of note, the available data are restricted to laboratory animals kept under SPF conditions and there are to the best of our knowledge no data available for humans and organisms that have longer life spans than mice and are exposed to real life challenges.

Second, as the authors also propose, engrafted macrophages are likely to undergo clonal proliferation in the brain if they originate from precursors that are recruited early on. However, this is not demonstrated in this study. Has BrdU labelling or any similar approach been used to prove that proliferation of exogenous macrophages in fact takes place in the brain parenchyma? We agree with the reviewer that engrafted cells likely spread in the host brain following proliferation and that this is an interesting aspect. That is why in our original submission we had addressed this issue and performed two independent experiments using genetically labelled cells: (1) Confetti-reporter encoded and (2) barcoded by lentiviral transduction. These approaches are similar, but arguably superior to BrdU labeling, as Confetti labeling provides in addition positional evidence to clonal proliferation within the brain parenchyma. BrdU labeling alone would not distinguish between proliferation in the BM niche and periphery (before engraftment), and local proliferation in the brain. The expansion of barcodes specific to engrafted macrophages also supports the findings in the Confetti labeling approach. We have modified the relevant text passages to better clarify how we concluded that clonal expansion of grafts takes place in the brain.

- The authors provide very little information about the actual location of engrafted cells compared to resident microglia in the context of blood vessels and brain ventricles. They mention concerning the Confetti study that „Seamless integration of engrafted cells into the endogenous microglial network was reflected by their Iba-1 expression, similar morphology and intercellular distances as compared host microglia”, but do not show that these cells display similar distribution to resident microglia or that intercellular distances are similar. Instead, on the images presented the cells appear in clusters in most brain areas and not too many of them can be seen in the cortex and the hippocampus compared to the numbers in the olfactory bulb and cerebellum, where the turnover of microglia is reportedly faster. This would argue against the real parenchymal integration of these cells. Previous BM transplantation studies have demonstrated the preferential proximity of blood-borne macrophages to ventricles and blood vessels, but it remained unclear whether changes in the brain microenvironment by irradiation itself played a role in later distribution of engrafting cells. Has any comparison between busulfan treatment and irradiation been performed in this regard (as for example suggested by Wilkinson et al., Mol Ther. 2013)?

In response to the reviewer's comment we have calculated intercellular distances of engrafted and host cells in brains of busulfan and TBI-conditioned mice. These data are now included as **Supplementary Figure 3G** in the revised manuscript.

We agree that side-by-side comparison of different conditioning protocols and detailed location analysis, such as the comprehensive one performed by Bigger and colleagues is important, but believe this is beyond the scope of the present study. Preferential engraftment of the cerebellum has been reported earlier (Priller et al. 2001) and we have included a reference to this study. As for the exact positioning of the cells we do believe we provide evidence for co-localization with host microglia.

- Many studies have used simple or sophisticated anatomical tools to discriminate different

activation stages of resident brain microglia in response to inflammatory challenges or disease. Have the authors tried to perform Sholl analysis or similar automated analysis to compare the branching characteristics and general morphology of microglia and engrafted macrophages based on Iba1 immunofluorescence after several months of engraftment? Such analysis would strengthen the statement regarding that engrafting macrophages adopt microglia characteristics. These data would be also very valuable for studies not able to discriminate resident microglia from exogenous macrophages based on genetic or fluorescent markers.

In response to the reviewer's comment we have performed a morphological analysis of engrafted and host cells in brains of busulfan and TBI-conditioned mice. This includes soma and cell areas (**Suppl. Fig. 3B, C**), numbers of processes and process lengths (**Suppl. Fig. 3D, E**) and maximal numbers of intersections (Sholl analysis) (**Suppl. Fig. 3F**).

For the parameters tested we do not see significant differences between engrafted and host brain macrophages.

- Similarly to recent papers by others, the present study shows that while host microglia and engrafted macrophages display significant transcriptome overlap (partially due to the impact of the brain microenvironment on the cells), both microglia and macrophages maintain their transcriptomic identity. However, it is also proposed in the paper that grafted CNS macrophages displayed an activation signature. From a future therapeutic point of view this issue is important as this data suggest that even several months after transplantation engrafting macrophages would not fully adapt to the brain microenvironment. The reasons for this is currently unclear, but the authors should further elaborate on this based on their transcriptomic data sets and discuss this issue in the manuscript.

In response to this reviewer's comment we now include in the revised manuscript a more detailed bio-informatic analysis. Specifically we added a GSEA analysis (**Suppl. Fig. 6A**) and an Ingenuity Pathway analysis (**Suppl. Fig. 9**), which both strengthen our notion that engrafted macrophages display an activation signature or are more poised to be active. In addition we have added a note on the similarity of engrafted macrophages to perivascular macrophages. Of note, we have added in the revised manuscript a figure that better highlights our FACS strategy to exclude a contamination of PVM when we sorted the samples (**Suppl. Fig. 4**)

Further points:

- It would be helpful to also show quantitative data about the number of CD45^{high} monocyte populations in the brain from graft1, graft2 and resident cells for the proper interpretation of the data presented in Fig.1.

In response to the reviewer's request we have now added an analysis of CD11b⁺ non-microglia cells in the brains of the 'tandem engrafted' chimeras (**Suppl. Fig. 1F**).

Of note, CD11b⁺ CD45^{hi} perivascular macrophages are sensitive to irradiation and hence more efficiently replaced by the graft than microglia (see f.i. also the newly added **Suppl. Fig. 4A, B**).

- Fig.1D. Graph supposedly shows Ly6C/G-negative cells as stated in the legend, unlike seen in the figure itself

This typo has been corrected in the revised draft. Cells are gated as shown in Fig. 1 C.

- Fig.1E would be more informative with the addition of a third channel showing all microglia visualized by immunofluorescence (Tmem119, P2y12, etc).

Unfortunately we are not able to comply with this request, since due to a freezer shut down we lost the blocks and hence cannot redo this immuno-histochemical analysis.

- In the genetic barcoding experiment the authors used busulfan instead of irradiation, hence no major BBB injury is expected to take place in these studies. However, it would be helpful if the authors would demonstrate this in the paper.
It has previously been established that busulfan conditioning allows the engraftment of the CNS by HSC-derived cells (Wilkinson et al 2013; Kierdorf et al 2013). We agree with the reviewer that the exact entry route would be interesting to explore further, but believe this is beyond the scope of this study.
- The level of significance between groups in the case of graphs presented in Fig.5C is unclear.
We apologize for having been unclear. The transcripts displayed show significant difference between at least one of the conditions. In response to the reviewer's comment and to be more specific we now have added significance between the different groups in **Fig 5C**.
- The quality of P2RY12 staining presented in Fig.7D is suboptimal, especially compared to the importance of the statement that resident P2Y12-positive microglia lack the Y-chromosome.
The P2RY12 staining is reproducibly established in the Prinz laboratory and its associated clinical pathology unit. It was performed on multiple individual brain samples from female patients who received BM transplants from male donors. We have been able to draw clear conclusions in another ongoing study. For better visualization of host and transplanted cells, we replaced the representative P2RY12 images for cortex and hippocampus in **Fig. 7D** with results from a recent staining, which reproduced earlier staining intensity.
- The text „Mean data are shown.” appears in duplicate in the Methods. Welch Two (tailed?) t-test also needs attention.
We apologize for these typos and have fixed them.

Reviewer #2

This is an interesting work, demonstrating that BM-derived cells that engraft the brain after irradiation or myeloablation are distinct from the resident microglia in their transcriptome, epigenome and response to immunological stimulus. The authors conclude that the engrafting cells, although take over the niche, do not acquire full microglia phenotype. The consequence for brain function of such engraftment is not demonstrated and it is therefore unclear whether the function of these new cells is different or similar to yolk-sac derived microglia.

While the work is interesting although mostly unsurprising, the major weakness is its novelty. Works over the last few months have demonstrated, using a more elegant than irradiation method (namely, deletion of M-CSF from microglia and thus rendering them incapable of proliferation and refilling the niche), identical results upon BM-derived cell engraftment or upon injection of different progenitors. These two works, although cited in the paper, are only briefly mentioned. The authors are encouraged to discuss these two works in more details and preferably compare their results to those in previous publication (including transcriptomic analyses of the engrafted cells).

Having more functional outcomes (behavioral aspects, electrophysiology, astrocyte physiology, etc) as a result of microglia replacement by cells of another origin, would make this work more exciting and innovative.

We agree with this reviewer that we were scooped and that some of our novelty was compromised.

However, we would like to point out that the paper by Kipnis and colleagues was published on line April 11th and the in our view more comprehensive study by Bennett et al. was published 1st of June, so our submission was very close to these two studies and our experimentation was of course performed well before these dates.

More importantly though, we have critical additions that substantiate and expand the findings, such as the epigenome analysis, the response to challenge and the analysis of patient brains.

We agree that the approaches used by Cronk and Bennett are elegant. However, irradiation and myeloablation employed in our study are arguably more straightforward and, as we show in our patient brain analysis, also clinically relevant.

In response to the reviewer's comment we performed a bioinformatics comparison of the data presented by Cronk et al. and Bennett et al studies and our data.

Many of key genes that display robust differential expression between engrafted macrophages and host microglia, including *Sall1*, *ApoE* and *Ms4a7* are reproduced in these three independent studies. We have added a respective Venn diagram analysis to the revised manuscript as **Suppl. Fig. 6B** and will include in the final submission the respective spread sheets. Differences between the gene lists are likely due to experimental details, incl. purity of sorted samples and distinct library production and RNAseq techniques used.

By analyzing their responses to peripheral LPS challenge we establish that brain macrophages and microglia are functionally distinct. We agree with the reviewer that it will be interesting to test in the future if the altered brain macrophages will contribute differentially to brain performance. We are currently designing such experiments, based on the differential expression profiles and LPS responses we observed. However, we believe this is beyond the scope of the present study.

Reviewer #3

I have read with interest the elegant work by Steffen Jung's group. Their conclusion that engrafted parenchymal brain macrophages differ from host microglia in transcriptome, epigenome and LPS responsiveness is convincing, and its significance is enhanced by supporting findings in human recipients of hemopoietic stem cell transplants. Overall, the epigenomic and transcriptomic data, on which I was asked to focus, are also robust and support the general conclusions of the paper. However, these analyses could be improved by integrating the RNASeq/ATACseq combination with targeted assessments of post-translational histone modifications and Chip-based approaches to specifically characterize regions/loci of interest. Indeed, as the authors acknowledge, "ATACseq does not discriminate between bound transcriptional activators and repressors". Therefore, some differentially expressed loci did not show epigenetic differences between engrafted parenchymal brain macrophages and host microglia. For some of those loci (and for the most compelling candidate loci that did show epigenetic differences), further characterization of histone marks and identification of transcription factors docking at sites of interest would highlight the regulatory landscape and would therefore be preferable to the purely bioinformatic approach currently adopted. Such studies would decisively deepen our understanding of the extent to which engrafted macrophages and host microglia diverge in regulatory terms.

We thank the reviewer for his supportive comment.

We of course agree that addition of Chip-based data could strengthen our conclusion on the epigenetic differences between the grafted and host macrophages and provide more in depth insights. However, inclusion of these data would result in a considerable delay of publication and in light of the recent studies by Kipnis and Barres we would like to avoid that.

Reviewers' comments:

Reviewer #1 (Remarks to the Author):

No further questions

Reviewer #2 (Remarks to the Author):

Unfortunately, the authors have not responded to my main critiques, such as functional assessment after microglia replacement. In my review of their responses to other reviewers, they also have not provided any experimental data. The authors argue that adding more details will further delay their publication as compared to that of Kipnis and Barres. This is an irrelevant comment, as the other works have already been published a few months ago. To publish their work, the authors need to provide some additional novelty.

Reviewer #3 (Remarks to the Author):

Dr. Jung's frustration at having been "scooped", as he puts it, by two recent publications is both palpable and understandable - novelty being a (the?) key criterion for getting our work published well. However, the solution to this problem is not (or not only) to point to commonalities between the recent publications and the work under review. Instead, I fully agree with Reviewer 2 in encouraging Dr. Jung and his team to discuss the previous works in detail and explicitly point to the data in their paper that are novel enough to advance the field. By the same token, I do not support Dr. Jung's refusal to engage in more extensive epigenetic studies on the grounds that this would delay publication. On the contrary, these studies might be part of what is needed to give this paper the edge that was taken away by the previous publications. Therefore, I recommend that this paper be further revised – at a minimum, in its text and analyses, and ideally also experimentally, by expanding the epigenetic characterizations along the lines discussed in my original comments.

Reviewer #2 (Remarks to the Author):

Unfortunately, the authors have not responded to my main critiques, such as functional assessment after microglia replacement. In my review of their responses to other reviewers, they also have not provided any experimental data. The authors argue that adding more details will further delay their publication as compared to that of Kipnis and Barres. This is an irrelevant comment, as the other works have already been published a few months ago. To publish their work, the authors need to provide some additional novelty.

Functional assays

We establish that microglia and BM-graft derived brain macrophages are functionally distinct in that they display distinct responses to the peripheral LPS challenge.

As stated in our point-by point reply, we agree with the reviewer 2 that additional functional assays will be important to assess the impact of our finding for brain physiology and pathophysiology.

Indeed, it is known in the community that BM chimeras respond different from non-irradiated animals in two common animal models for CNS pathology, i.e. experimental autoimmune encephalomyelitis (EAE) and cuprizone-induced de-myelination. The underlying reason for these discrepancies remains however unknown and could be due to altered BBB permeability and hence altered monocyte influx, but also to the phenomenon we describe. To conclusively answer this point, we believe one has to devise a model that differentiates between acute monocyte infiltrates and the brain macrophage population that entered the brain during the generation of the chimeras. We are working on establishing such a system using a combination of fate mapping and 'ribotagging' with the CX₃CR1^{CreER} model (Haimon et al., *Nat Immunol* 2018). Implementation of such a complex experimental system is however time consuming and we will not be able to have these data before the spring 2019. Likewise, we believe *in vitro* assays with isolated BM-derived CNS macrophages and microglia are of limited value since the Barres group has convincingly shown that *in vitro* cultured brain microglia rapidly lose their identity (Bohlen et al. 2017), and indeed acquire a signature closer to the BM-derived brain macrophages. We abstained from *in vitro* cultures.

In response to the request of the reviewers we nevertheless decided to perform such experimentation. Specifically, we decided to test the induction of reactive oxygen species (ROS) by an acute LPS challenge. Our RNAseq data showed distinct expression between the graft and host cells, which we had hoped to pick up in the *in vitro* assay (**Fig 1A**). However, when we analyzed the brain macrophages of the chimeras, differentiating graft and host by the CD45 marker, we did not observe any conclusive difference in the induction (**Fig 1B, C**). As outlined above, these *in vitro* assays are likely compromised by isolation techniques and rapid transcriptome changes, as reported by Bohlen and colleagues and ourselves (*Haimon et al 2018*). We would hence like to abstain from including these results in the manuscript and reserve them for a more comprehensive in depth follow up study, including *in vivo* experimentation.

Figure 1. Analysis of ROS production by engrafted macrophages and host microglia, 4 hrs after in vitro LPS challenge.

- A) RNAseq expression data of *Cybb* gene encoding super-oxide generating Nox2 enzyme which forms reactive oxygen species (ROS) and iNOS/ NOS2.
- B) Gating strategy for analysis of grafted and host macrophages
- C) Summary of ROS production, control n=3, LPS n= 5 genes associated with ROS production

Reviewer #3 (Remarks to the Author):

Dr. Jung's frustration at having been "scooped", as he puts it, by two recent publications is both palpable and understandable - novelty being a (the?) key criterion for getting our work published well. However, the solution to this problem is not (or not only) to point to commonalities between the recent publications and the work under review. Instead, I fully agree with Reviewer 2 in encouraging Dr. Jung and his team to discuss the previous works in detail and explicitly point to the data in their paper that are novel enough to advance the field. By the same token, I do not support Dr. Jung's refusal to engage in more extensive epigenetic studies on the grounds that this would delay publication. On the contrary, these studies might be part of what is needed to give this paper the edge that was taken away by the previous publications. Therefore, I recommend that this paper be further revised – at a minimum, in its text and analyses, and ideally also experimentally, by expanding the epigenetic characterizations along the lines discussed in my original comments.

Additional Epigenetic Analysis

As outlined in our last response, we agree with reviewer 3 that future additional epigenome analysis is warranted to better understand the differences between the two macrophage populations, including ChIPseq and DNA methylation analyses. We have launched such efforts, but believe they are beyond the scope of this study.

To nevertheless substantiate our conclusion that engrafted and host macrophages differ we include in Figure 6 of the revised manuscript both RNAseq and ATACseq data for another gene, *Spp1*, encoding osteopontin, whose expression has been linked to disease states. The differential induction we observe could have implications for brain pathologies. We discuss this now in a new paragraph in the discussion of the revised manuscript (see below). Also, the *Spp1* locus displayed a nice differential ATAC seq differential and we hence added it to Figure 6 (**Fig 2**).

Figure 2 (Fig 6 in revised manuscript). LPS Challenge induced alterations in the *Marco* locus (C) and *Spp1* locus (D). Normalized sequence reads mRNA in engrafted cells and host microglia isolated from LPS challenged and unchallenged BM chimeras; Normalized ATACseq profiles of loci with enlarged areas highlighting induced ATACseq peaks and predicted motifs.

Discussion of recent studies by the Kipnis and Barres laboratories.

We include in our discussion a dedicated paragraph discussing the two recently published works in more detail and explicitly point to the data in our paper that advance the field. We also include a comparison of the data of the three studies. As outlined before our study considerably extends these reported findings by including **(1)** the tandem transfers and clonality analysis of grafted cells, **(2)** a comparison of both transcriptomes and epigenomes of grafted and host cells **(3)** a comparison of responses to LPS challenge and **(4)** an analysis of human brain tissue of patients that received HSC transplant. We feel that these significant additions over the studies by the Kipnis and Barres laboratories, which provide significant novelty, were not fully appreciated by the reviewers. Finally, to our understanding NComms had agreed to send this out in full knowledge of the “scooping” implying that you were content with the extent of the advance by our study. We had clearly stated the situation in our cover letter. “Scooping” should hence not be the factor in the decision-making.